# TRACER: Trajectory Risk Aggregation for Critical Episodes in Agentic Reasoning

**Sina Tayebati** [1]  **Divake Kumar** [1]  **Nastaran Darabi** [1]  **Davide Ettori** [1]  **Ranganath Krishnan** [2]
**Amit Ranjan Trivedi** [1]

## Abstract

Estimating uncertainty for *AI agents* in real-world multi-turn tool-using interaction with humans is difficult because failures are often triggered by sparse *critical episodes* (e.g., looping, incoherent tool use, or user-agent miscoordination) even when local generation appears confident. Existing uncertainty proxies focus on single-shot text generation and therefore miss these trajectory-level breakdown signals. We introduce **TRACER**, a trajectory-level uncertainty metric for dual-control Tool-Agent-User interaction. TRACER combines content-aware surprisal with situational-awareness signals, semantic and lexical repetition, and tool-grounded coherence gaps, and aggregates them using a tail-focused risk functional with a MAX-composite step risk to surface decisive anomalies. We evaluate TRACER on $\tau^2$-bench (Barres et al., 2025) by predicting task failure and selective task execution. To this end, TRACER improves AUROC by up to 37.1% and AUARC by up to 55% over baselines, enabling earlier and more accurate detection of uncertainty in complex conversational tool-use settings. Our codebase is available at: *https://github.com/sinatayebati/agent-tracer*.

## 1. Introduction

Large language model (LLM) agents are increasingly deployed in interactive, open-ended settings that require sustained multi-turn dialogue, external tool use, and coordination with a user. In such agentic environments, failures are rarely caused by isolated token-level errors. Instead, breakdowns arise from trajectory-level phenomena: task drift, repetitive behavior, misinterpretation of tool outputs, or misalignment with the user's evolving intent. Crucially, these failures often occur despite fluent and locally confident generation, exposing a gap between standard uncertainty proxies and the realities of multi-turn, tool-using decisions.

Estimating agent uncertainty must therefore account for the dynamics of conversational trajectories. Most existing uncertainty measures operate at the token or sequence level, e.g., predictive entropy, self-consistency, or likelihood-based surrogates, and are typically evaluated in static, single-shot settings (Malinin & Gales, 2021; Kuhn et al., 2023; Wang et al., 2023; Jiang et al., 2020). In task-oriented interaction, however, success depends on maintaining situation awareness across turns: tracking constraints, updating beliefs from observations, and adapting to user feedback and tool outputs. Failures are frequently driven by sparse critical episodes, such as looping, incoherent action–outcome alignment, or user-agent coordination collapse, rather than uniformly elevated uncertainty throughout the trajectory (Shinn et al., 2023; Liu et al., 2024; Wang et al., 2024; Tan et al., 2024). As a result, global averages of token uncertainty systematically underweight the segments that determine success or failure.

We introduce **TRACER** (Trajectory Risk Aggregation for Critical Episodes in agentic Reasoning), a trajectory-level uncertainty metric for dual-control agentic environments in which both the agent and the user are stochastic and the agent reasons, communicates, and uses tools, as shown at Figure 1. TRACER extends prior uncertainty propagation approaches by (i) using content-aware normalized surprisal to emphasize epistemically meaningful tokens, (ii) incorporating situational-awareness indicators that capture looping and coherence gaps, and (iii) aggregating step risks with a tail-focused risk functional that emphasizes worst-case dialogue segments rather than average behavior. This design directly targets the dominant failure regime such as sparse but decisive uncertainty episodes that derail coherence.

Beyond introducing the metric, we provide formal justification for TRACER as a principled trajectory-level risk measure. We show that the core token uncertainty term admits an information-theoretic decomposition into intrinsic

[1]Department of Electrical and Computer Engineering, University of Illinois Chicago, IL, USA [2]Capital One, AI Labs, TX, USA. Correspondence to: Sina Tayebati <stayeb3@uic.edu>.

*Proceedings of the $43^{rd}$ International Conference on Machine Learning*, Seoul, South Korea. PMLR 306, 2026. Copyright 2026 by the author(s).

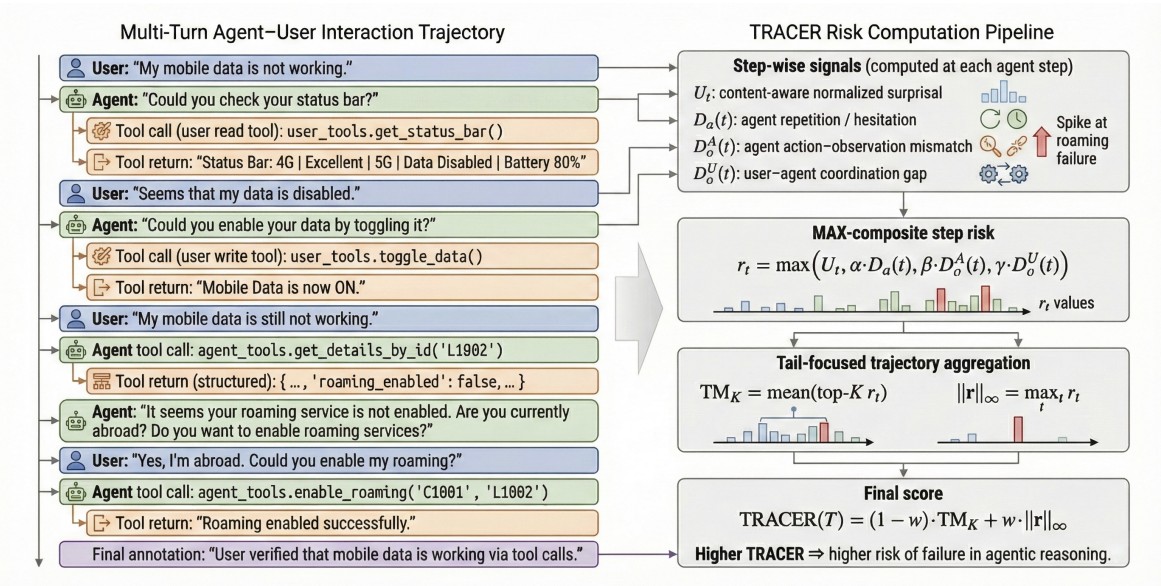

*Figure 1.* Overview of TRACER for trajectory-level uncertainty estimation in agentic reasoning. **Left:** A multi-turn Agent–User interaction with tool calls and delayed failure resolution. **Right:** At each agent step $t$, content-aware surprisal $U_t$, agent repetition $D_a(t)$, action–observation mismatch $D_o^A(t)$, and user–agent coordination gap $D_o^U(t)$ are computed and combined via a MAX-composite step risk, $r_t = \max(U_t, \alpha D_a(t), \beta D_o^A(t), \gamma D_o^U(t))$. Trajectory risk is obtained through tail-focused aggregation (top-$K$ mean and $\ell_\infty$ norm).

content uncertainty and epistemic mismatch, and that the aggregation defines a coherent tail-risk functional with stability guarantees. Under a sparse-hazard breakdown model, we establish conditions under which TRACER upper-bounds trajectory-level failure probability, grounding it as a theoretically motivated proxy for uncertainty-driven breakdown in multi-turn conversations.

## 2. Related Work

**Uncertainty estimation in language models.** Uncertainty quantification for LLMs has gained attention through predictive entropy, calibration, and Bayesian or ensemble-inspired methods (Malinin & Gales, 2021; Zhao et al., 2021). Common proxies include token-level entropy and surprisal, sequence log-likelihood, temperature-scaled confidence, and sampling-based techniques such as self-consistency or generation variance (Kuhn et al., 2023; Wang et al., 2023). While effective in static tasks, these measures are often insufficient in agentic settings where failure depends on long-horizon state tracking, tool interaction, and user feedback. Token-level confidence can remain high even as an agent enters repetitive loops or misuses tools, motivating uncertainty metrics that incorporate behavioral signals beyond likelihood (Shinn et al., 2023; Lin et al., 2023).

**Calibration, selective prediction, and risk measures.** Related work studies calibration and selective prediction, including confidence-based abstention, conformal prediction

(Tayebati et al., 2025), and risk control under distribution shift (Angelopoulos & Bates, 2021). These approaches provide guarantees for deferral by mapping a scalar uncertainty score to a selective decision rule (Geirhos et al., 2020). However, extending them to multi-turn agentic interaction is nontrivial, since the object of interest is a trajectory rather than a single prediction, and errors can be sparse yet catastrophic. TRACER connects to this literature by constructing a trajectory-level score that behaves as a coherent tail-risk measure, emphasizing worst-case dialogue segments rather than average uncertainty (Chow et al., 2017). Further, studies such as Rho-1 by (Lin et al.) focus on reweighting or selecting tokens to improve training dynamics and loss shaping, whereas TRACER extends this philosophy to inference-time observability for risk estimation. Specifically, we employ structural filtering of highly predictable tokens to mitigate the dilution of epistemic uncertainty during open-ended generation.

**Situation awareness and failure modes in agentic systems.** Agentic evaluation emphasizes long-horizon planning, tool invocation, and state maintenance across dialogue (Jimenez et al., 2024). Empirical studies identify recurring failure modes such as infinite loops, instruction drift, hallucinated tool outputs, and brittle user coordination (Liu et al., 2024; Wang et al., 2024). These failures are often structural rather than semantic, manifesting as repeated actions, mismatched observations, and incoherent turn-to-turn transitions. TRACER operationalizes these phe-

nomena through explicit situational-awareness indicators for repetition and coherence gaps, complementing token-level uncertainty with trajectory-diagnostics (Yao et al., 2023).

**Tool-using agents and multi-turn benchmarks.** A growing set of benchmarks evaluates tool use, web navigation, and multi-step reasoning in interactive environments (Zhou et al., 2023; Jimenez et al., 2024). These settings highlight that observation feedback is external and may be delayed or noisy, and that success depends on aligning actions with tool outputs over multiple turns. Uncertainty estimation must therefore integrate action–outcome coherence and user–agent alignment while remaining sensitive to rare but decisive breakdown episodes. TRACER is designed for these settings by modeling dialogue as a dual-control trajectory and scoring risk at the step level before tail-focused aggregation (Dai et al., 2024; Chow et al., 2017).

**Contributions and novelty.** TRACER introduces a trajectory-level uncertainty metric for real-world agentic interaction. We propose step-wise signals that go beyond token-level likelihood to capture uncertainty mechanisms specific to multi-turn, tool-using dialogue: (i) a normalized surprisal term emphasizing epistemically meaningful content; (ii) *lexical and semantic repetition signals* that detect degenerate looping and stagnation missed by standard proxies; and (iii) *situation-awareness gap* indicators, $\mathbf{D_o}$ and $\mathbf{D_a}$, measuring inconsistencies between observed tool or environment feedback and subsequent agent statements ($D_o$), and between agent actions and implied task constraints or outcomes ($D_a$). Both indicators are trajectory-consistent, defined over multi-turn context windows, and reflect the agent's ability to maintain coherent belief updates and action–outcome alignment under external feedback. These signals are aggregated using a *tail-focused coherent risk functional* that emphasizes critical episodes rather than average behavior. We further provide theoretical support showing that TRACER is a principled trajectory-risk measure with stability guarantees and conditions under which it upperbounds uncertainty-driven breakdown probability.

## 3. Methodology

We develop a mathematically grounded framework for quantifying *trajectory-level uncertainty risk* of conversational agents in the dual-control setting of $\tau^2$-Bench, where both the agent and the user follow stochastic policies. Our method builds on prior uncertainty propagation and situation-awareness based approaches to yield a tail-risk formulation suitable for dual-control interaction. Unlike prior uncertainty metrics that average token-level uncertainty over an interaction, we model failures in multi-turn, tool-using dialogue as arising from *critical episodes*, such as looping, incoherent tool use, or user-agent coordination collapse, which can occur even when token uncertainty is low.

TRACER therefore integrates (i) content-aware token surprisal, (ii) situational-awareness indicators for repetition and coherence gaps, and (iii) a tail-focused aggregation that emphasizes the most failure-relevant segments of a trajectory. Formal statements and proofs are provided in Appendix A.

### 3.1. Dual-Control Trajectory Model

Let $\mathcal{A} \triangleq \{\text{Agent}, \text{User}\}$ denote the set of actors. A dialogue trajectory of length $N$ is

$$\mathcal{T} \triangleq \left\{ (a_t, x_t, o_t) \right\}_{t=1}^{N}, \qquad (1)$$

where $a_t \in \mathcal{A}$ identifies the actor at step $t$, $x_t \in \mathcal{X}$ is the emitted event (utterance or structured action), and $o_t \in \mathcal{O}$ is the resulting observation returned by the environment, including tool outputs, simulator responses, or counterparty messages. Each event is associated with a textual realization $z(x_t) \in \Sigma^\star$ and, when applicable, each observation with $z(o_t) \in \Sigma^\star$. TRACER maps trajectories to scores:

$$\text{TRACER} : \bigcup_{N \geq 1} (\mathcal{A} \times \mathcal{X} \times \mathcal{O})^N \to \mathbb{R}_{\geq 0}, \qquad (2)$$

where larger values indicate a higher propensity for uncertainty-driven breakdown.

### 3.2. Stage I: Content-Aware Normalized Surprisal

Consider a generative step $t$ where the actor $a_t$ produces a token sequence

$$y_t \triangleq (w_{t,1}, w_{t,2}, \ldots, w_{t,L_t}) \in \mathcal{V}^{L_t}, \qquad (3)$$

conditioned on a context $\mathcal{C}_t$ (dialogue state, tool traces, and prior messages). Let

$$p_t(w_{t,j}) \triangleq \mathbb{P}(w_{t,j} \mid w_{t,<j}, \mathcal{C}_t) \qquad (4)$$

denote the predictive probability of token $w_{t,j}$. Averaging token-level surprisal over all tokens can be dominated by structural and function-word tokens, diluting the uncertainty signal. We thus only aggregate on *content-bearing* tokens.

**Content-bearing token predicate.** Fix (i) a stop-word set $\mathcal{S}_{\text{stop}} \subset \mathcal{V}$, (ii) a predicate $\text{Num} : \mathcal{V} \to \{0, 1\}$ indicating whether a token is purely numeric (up to punctuation), and (iii) a probability threshold $\pi_0 \in (0, 1)$ to suppress highly predictable structural tokens. Define

$$\chi(w_{t,j}) \triangleq \mathbf{1}\{p_t(w_{t,j}) \leq \pi_0\} \cdot \mathbf{1}\{w_{t,j} \notin \mathcal{S}_{\text{stop}}\} \\ \cdot \mathbf{1}\{\text{Num}(w_{t,j}) = 0\}. \qquad (5)$$

$$\text{Let} \quad \mathcal{I}_t \triangleq \{j \in \{1, \ldots, L_t\} : \chi(w_{t,j}) = 1\}. \qquad (6)$$

**Filtered normalized surprisal.** We define the step-wise uncertainty signal

$$U_t \triangleq \begin{cases} \dfrac{1}{|\mathcal{I}_t|} \sum_{j \in \mathcal{I}_t} \left[ -\log p_t(w_{t,j}) \right], & \text{if } |\mathcal{I}_t| > 0, \\ \epsilon, & \text{if } |\mathcal{I}_t| = 0, \end{cases} \qquad (7)$$

for a small $\epsilon > 0$. In $\tau^2$-Bench, both agent and user turns may be language models; whenever token probabilities are available, (7) applies symmetrically to either actor.

### 3.3. Stage II: Situational Awareness Indicators

Token-level uncertainty alone cannot detect key agentic failure modes, particularly the *false-confidence* regime, where $U_t$ is small despite looping or incoherent tool use. We introduce explicit indicators for repetition and coherence.

#### 3.3.1. HYBRID LOCAL REPETITION FUNCTIONAL

Let $\varphi : \Sigma^\star \to \mathbb{R}^d$ denote an embedding map. Define cosine similarity

$$\mathrm{sim}_{\mathrm{sem}}(u,v) \triangleq \frac{\langle \varphi(u), \varphi(v) \rangle}{\|\varphi(u)\|_2 \, \|\varphi(v)\|_2}, \qquad (8)$$

with value 0 if either vector is zero. Let $\mathrm{ContTok}(u) \subset \mathcal{V}$ denote the set of content-bearing tokens extracted from $u$ using the same stop-word and numeric filters as in (5). Define the Jaccard overlap

$$\mathrm{sim}_{\mathrm{lex}}(u,v) \triangleq \frac{|\mathrm{ContTok}(u) \cap \mathrm{ContTok}(v)|}{|\mathrm{ContTok}(u) \cup \mathrm{ContTok}(v)|}, \qquad (9)$$

with $\mathrm{sim}_{\mathrm{lex}}(u,v) = 0$ if the denominator is zero.

Fix a local window size $m \in \mathbb{N}$. For an agent turn at time $t$ (i.e., $a_t = \mathrm{Agent}$), define

$$\begin{aligned} \mathcal{H}_{\mathrm{A}}(t) &\triangleq \{\, t' < t : a_{t'} = \mathrm{Agent} \,\}, \\ \mathcal{W}(t) &\triangleq \{\, t' \in \mathcal{H}_{\mathrm{A}}(t) : t - m \le t' < t \,\}. \end{aligned} \qquad (10)$$

Let $u_t \triangleq z(x_t)$. The hybrid repetition score is

$$\begin{aligned} D_{\mathrm{rep}}(t) \triangleq \max_{t' \in \mathcal{W}(t)} \Big[ &\mathrm{sim}_{\mathrm{sem}}(u_t, u_{t'}) \\ &\cdot \mathrm{sim}_{\mathrm{lex}}(u_t, u_{t'}) \Big], \end{aligned} \qquad (11)$$

with $D_{\mathrm{rep}}(t) = 0$ if $\mathcal{W}(t) = \emptyset$. The product implements a soft conjunction: true loops exhibit both high semantic similarity and high lexical overlap, while valid enumeration typically preserves semantic similarity but reduces lexical overlap due to entity changes.

**Notation.** To align with the action-side drift or repetition signal used in our implementation, we also write

$$D_a(t) \triangleq D_{\mathrm{rep}}(t), \qquad (12)$$

and use $D_a(t)$ and $D_{\mathrm{rep}}(t)$ interchangeably.

#### 3.3.2. INFERENCE-GAP (COHERENCE) FUNCTIONALS

We quantify semantic incoherence between actions and outcomes, as well as between agent messages and user responses. Define semantic distance

$$\delta(u,v) \triangleq 1 - \mathrm{sim}_{\mathrm{sem}}(u,v). \qquad (13)$$

**Agent coherence gap.** Let $\mathcal{T}_{\mathrm{tool}} \subset \{1, \dots, N\}$ denote indices at which the agent emits an action whose semantics should align with the resulting observation, such as tool usage. For $t \in \mathcal{T}_{\mathrm{tool}}$, define

$$D_o^{\mathrm{A}}(t) \triangleq \delta\big(z(x_t), z(o_t)\big). \qquad (14)$$

**User coherence (coordination) gap.** For a user turn at time $t$ with $a_{t-1} = \mathrm{Agent}$ and $a_t = \mathrm{User}$, define

$$D_o^{\mathrm{U}}(t) \triangleq \delta\big(z(x_{t-1}), z(x_t)\big). \qquad (15)$$

### 3.4. Stage III: MAX-Composite Step Risk and Tail Aggregation

#### 3.4.1. MAX-COMPOSITE STEP RISK

Let $U_t \in \mathbb{R}_{\ge 0}$ denote the content-aware normalized surprisal at step $t$, and let $D_{\mathrm{rep}}(t) \in [0,1]$, $D_o^{\mathrm{A}}(t) \in [0,2]$, and $D_o^{\mathrm{U}}(t) \in [0,2]$ denote the repetition, agent-coherence, and user-coherence indicators defined in Sec. 3.3.1–3.3.2. We form the following nonnegative component risks:

$$\begin{aligned} r_t^U &\triangleq U_t, & (16) \\ r_t^{\mathrm{rep}} &\triangleq \alpha\, D_{\mathrm{rep}}(t), & (17) \\ r_t^{\mathrm{A}} &\triangleq \beta\, D_o^{\mathrm{A}}(t)\, \mathbf{1}\{a_t = \mathrm{Agent}\}, & (18) \\ r_t^{\mathrm{Uo}} &\triangleq \gamma\, D_o^{\mathrm{U}}(t)\, \mathbf{1}\{a_t = \mathrm{User}\}, & (19) \end{aligned}$$

where $\alpha, \beta, \gamma \ge 0$ are scalar weights and $\mathbf{1}\{\cdot\}$ denotes the indicator function. The *MAX-composite* step risk is the pointwise maximum of these components:

$$r_t \triangleq \max\{r_t^U, \, r_t^{\mathrm{rep}}, \, r_t^{\mathrm{A}}, \, r_t^{\mathrm{Uo}}\}. \qquad (20)$$

Equivalently, letting $\mathbf{c}_t \triangleq (r_t^U, r_t^{\mathrm{rep}}, r_t^{\mathrm{A}}, r_t^{\mathrm{Uo}}) \in \mathbb{R}_{\ge 0}^4$, we have $r_t = \|\mathbf{c}_t\|_\infty$. This aggregation emphasizes the dominant failure signal at each step and avoids diluting sharp breakdown evidence by summing weaker components.

#### 3.4.2. TAIL MEAN AND WORST-CASE RISK

Let $\mathbf{r}(\mathcal{T}) \triangleq (r_1, \dots, r_N) \in \mathbb{R}_{\ge 0}^N$ and $r_{(1)} \ge r_{(2)} \ge \cdots \ge r_{(N)}$ denote the descending order statistics of $\mathbf{r}$. Fix a tail fraction $k \in (0,1]$ and define

$$K \triangleq \max\{1, \lfloor kN \rfloor\}. \qquad (21)$$

Define the empirical top-$k$ tail mean

$$\mathrm{TM}_k(\mathbf{r}) \triangleq \frac{1}{K} \sum_{i=1}^{K} r_{(i)}. \qquad (22)$$

Also define $\|\mathbf{r}\|_\infty \triangleq r_{(1)}$.

### 3.4.3. TRACER TRAJECTORY SCORE

TRACER convexly combines tail mean and worst-case risk:

$$\text{TRACER}_{\boldsymbol{\theta}}(\mathcal{T}) \triangleq (1 - w)\,\text{TM}_k(\mathbf{r}(\mathcal{T}))\ +\ w\,\|\mathbf{r}(\mathcal{T})\|_\infty,$$
$$\boldsymbol{\theta} \triangleq (\alpha, \beta, \gamma, k, w), \qquad w \in [0, 1].$$
$$(23)$$

The weight $w$ interpolates between chronic uncertainty patterns (captured by $\text{TM}_k$) and acute catastrophic breakdowns (captured by the maximum).

## 3.5. Risk Decomposition and Failure Control

We establish that TRACER is a mathematically principled trajectory-level uncertainty metric for multi-turn, task-oriented, dual-control dialogue. Our results characterize (i) the information-theoretic meaning of the content-aware surprisal signal, (ii) coherence and stability of the tail-risk aggregation, and (iii) conditions under which TRACER upper-bounds trajectory-level breakdown probability when failures arise from sparse critical episodes.

**Probabilistic setup.** Let $(\Omega, \mathcal{F}, \mathbb{P})$ support the dialogue trajectory $\mathcal{T} = \{(a_t, x_t, o_t)\}_{t=1}^N$ induced by the environment and stochastic agent and user policies. Let $\{\mathcal{F}_t\}_{t=0}^N$ be the filtration generated by the partial history up to step $t$. We model uncertainty via a possibly latent stepwise random variable $C_t \in [0, 1]$, measurable w.r.t. $\mathcal{F}_t$, representing the presence of a failure-inducing condition (e.g., looping, incoherent tool use, or user-agent miscoordination). The breakdown event is

$$B \triangleq \left\{ \max_{t \in \{1, \dots, N\}} C_t > 0 \right\},$$

formalizing that a single critical episode can derail an otherwise coherent interaction.

**Information-theoretic interpretation of content-aware surprisal.** For a generative step $t$, let $Q_{t,j}(\cdot) = \mathbb{P}(W_{t,j} = \cdot \mid W_{t,<j}, C_t)$ denote the true conditional token distribution and let $P_{t,j}(\cdot)$ denote the model distribution. The content-filtered cross-entropy

$$H_t^{\text{cont}}(Q, P) = \frac{1}{|\mathcal{I}_t|} \sum_{j \in \mathcal{I}_t} \mathbb{E}_{W \sim Q_{t,j}}[-\log P_{t,j}(W)]$$

admits the decomposition

$$H_t^{\text{cont}}(Q, P) = \frac{1}{|\mathcal{I}_t|} \sum_{j \in \mathcal{I}_t} H(Q_{t,j}) + \frac{1}{|\mathcal{I}_t|} \sum_{j \in \mathcal{I}_t} \text{KL}(Q_{t,j} \| P_{t,j}),$$

separating intrinsic (aleatoric) content uncertainty from epistemic mismatch. The empirical statistic $U_t$ used by TRACER is an unbiased estimator of $H_t^{\text{cont}}(Q, P)$ conditional on the step context. This justifies $U_t$ as a content-focused uncertainty proxy that is not diluted by predictable

structural tokens. When token probabilities are unavailable (e.g., black-box user simulators), TRACER omits $U_t$ and operates on situational-awareness signals only; all subsequent guarantees continue to hold with $r_t^U = 0$.

**Coherent and stable tail-risk aggregation.** Let $\mathbf{r} = (r_1, \dots, r_N)$ denote the MAX-composite step risks. TRACER aggregates risk via

$$\rho_{k,w}(\mathbf{r}) \triangleq (1 - w)\,\text{TM}_k(\mathbf{r}) + w\,\|\mathbf{r}\|_\infty,$$

where $\text{TM}_k$ is the empirical top-$k$ tail mean (CVaR). Since $\text{TM}_k$ is a spectral risk measure and $\|\cdot\|_\infty$ is a norm, $\rho_{k,w}$ is a coherent risk functional: it is monotone, translation invariant, positively homogeneous, and subadditive. Moreover, $\rho_{k,w}$ is 1-Lipschitz under $\ell_\infty$, ensuring robustness to local perturbations in step-level uncertainty signals. TRACER is monotone in each component signal: increasing any $U_t$, $D_a(t)$, or $D_o(t)$ cannot decrease the trajectory risk score.

**Failure-risk control under sparse hazards.** Define the conditional hazard $\lambda_t = \mathbb{P}(C_t > 0 \mid \mathcal{F}_{t-1})$. By a union bound,

$$\mathbb{P}(B) \leq \mathbb{E}\left[\sum_{t=1}^N \lambda_t\right].$$

We assume (i) *risk-dominates-hazard*: there exists $c > 0$ such that $\lambda_t \leq \min\{1, c\,r_t\}$ for all $t$, and (ii) *tail sparsity*: for $K = \max\{1, \lfloor kN \rfloor\}$, there exists $\eta \geq 0$ such that,

$$\sum_{i=K+1}^N r_{(i)} \leq \eta.$$

Under these conditions,

$$\mathbb{P}(B) \ \leq\ c\,K\,\mathbb{E}[\text{TM}_k(\mathbf{r})]\ +\ c\,\eta. \qquad (24)$$

Since $\text{TRACER}(\mathcal{T}) \geq (1 - w)\text{TM}_k(\mathbf{r})$ for $w < 1$, we further obtain

$$\mathbb{P}(B) \ \leq\ \frac{cK}{1 - w}\,\mathbb{E}[\text{TRACER}(\mathcal{T})]\ +\ c\,\eta.$$

The bound is conservative by construction, relying on a union bound and worst-case dominance, but is sufficient to justify tail-focused aggregation under sparse episodes.

**Dual-control attribution.** Because exactly one actor acts per step, the MAX construction yields a clean actor-wise decomposition of step risks. By subadditivity of $\rho_{k,w}$,

$$\text{TRACER}(\mathcal{T}) \leq \text{TRACER}_{\text{Agent}}(\mathcal{T}) + \text{TRACER}_{\text{User}}(\mathcal{T}),$$

enabling attribution of breakdown risk to agent-driven versus user-driven uncertainty without double counting. Unlike additive aggregation, the MAX step risk preserves sensitivity to single decisive failures without requiring calibration across heterogeneous uncertainty signals.

*Table 1.* Key statistics for the $\tau^2$-bench domains.

| | Retail | Airline | Telecom |
|---|---|---|---|
| **Agent DB** | 500U, 50P, 1kO | 500U, 300F, 2kRes | 5P, 9L, 4C |
| **Agent Tools** | 7W, 6Rd | 6W, 6Rd | 6W, 7Rd |
| **User Tools** | – | – | 15W, 15Rd |
| **Tasks** | 115 | 50 | 114 |

*Notation:* U=users, P=products or plans, F=flights, O=orders, Res=reservations, L=lines, C=customers; W=write, Rd=read.

**Computational Complexity.** Let $L_t$ denote the token length at step $t$, and let $c_\varphi(z)$ denote the cost of computing the embedding $\varphi(z)$. Computing the content-aware surprisal (7) over a trajectory requires evaluating token log-probabilities and costs $\mathcal{O}\left(\sum_{t=1}^{N} L_t\right)$, assuming token probabilities are available from the generating model.

Situational-awareness indicators require embedding-based comparisons. With embedding caching and a local window of size $m$, the dominant embedding cost is

$$\mathcal{O}\left(\sum_{t=1}^{N} c_\varphi(z(x_t)) + \sum_{t=1}^{N} c_\varphi(z(o_t))\right),$$

while all similarity computations within the window incur only $\mathcal{O}(mN)$ arithmetic operations and are negligible relative to embedding cost.

The MAX-composite step risk (20) is computed in constant time per step, contributing $\mathcal{O}(N)$ total cost. Tail aggregation requires sorting the step risks $\{r_t\}_{t=1}^{N}$, incurring $\mathcal{O}(N \log N)$ time, or $\mathcal{O}(N)$ expected time using selection-based top-$K$ algorithms. Overall, the complexity is

$$\mathcal{O}\left(\sum_{t=1}^{N} L_t + \sum_{t=1}^{N} c_\varphi(z(x_t)) + \sum_{t=1}^{N} c_\varphi(z(o_t)) + N \log N\right),$$

with memory complexity linear in $N$. For parameter identification over a finite grid of $\boldsymbol{\theta}$ values, step-level quantities $(U_t, D_a, D_o^{\mathrm{A}}, D_o^{\mathrm{U}})$ are precomputed once and reused. As a result, evaluation across candidate parameters is dominated by repeated tail aggregation and lightweight arithmetic, making the marginal cost per configuration $\mathcal{O}(N \log N)$.

## 4. Results and Discussions

### 4.1. Evaluation Protocol

We evaluate TRACER on the $\tau^2$-bench dual-control environment (Barres et al., 2025), a testbed for multi-turn *Tool–Agent–User* (TAU) interactions in which both the conversational agent and a simulated user act on a shared, partially observed world via natural language and tool calls. Interactions are modeled as a two-player Dec-POMDP (Oliehoek & Amato, 2016), capturing realistic scenarios, such as troubleshooting, that require goal-directed tool use. Furthermore, to solidify our empirical evaluation, we continue our

experiments with two additional benchmarks, ToolHop (Ye et al., 2025) and ComplexFuncBench (Zhong et al., 2025).

**Dec-POMDP formalism.** The interaction is defined by $(\mathcal{S}, \{\mathcal{A}_i\}, \{\mathcal{O}_i\}, \mathcal{T}, \mathcal{R}, \mathcal{U}, \mathcal{M})$ with players $i \in \{\texttt{agent}, \texttt{user}\}$. At each turn, exactly one player acts by emitting a message in $\mathcal{M}$ or issuing a tool call, after which the environment transitions and returns player-specific observations. This structure is central to TRACER, as uncertainty in TAU settings is interactional, arising from partial observability, tool-mediated state changes, and user-agent miscoordination rather than uncertainty about a final answer.

**Domains and datasets.** As summarized in Table 1, we evaluate on all verified $\tau^2$-bench domains, including airline, retail, and telecom (Barres et al., 2025). In each domain, the agent accesses database-style tools (e.g., retrieval and verification), while the user accesses device-level tools (e.g., toggling phone settings), inducing decentralized control and requiring explicit coordination (Qin et al., 2024). Tasks are programmatically generated and automatically verifiable; in telecom, success is determined via assertion checks on the final world state. We follow the benchmark's task definitions and success criteria.

**Agent setting.** We follow the standard $\tau^2$-bench agent implementation protocol (Barres et al., 2025). All LLM calls to `gemini-2.5-flash`, `gemini-2.5-pro`, `gpt-4.1-mini`, and `gpt-5.4-mini` are executed via the LiteLLM framework. Both the agent and the user simulator are implemented as function-calling agents using the OpenAI tools format (OpenAI, 2023). The agent prompt includes generic guidelines and domain-specific policies, while the user prompt includes generic constraints and task-specific instructions (Yao et al., 2023). All tasks are run with temperature 0 and log probabilities enabled, ensuring stable and comparable estimates.

**Problem formulation with TRACER.** We formulate uncertainty estimation as a *failure prediction* problem over TAU trajectories. For each episode $e$, let $y_e \in \{0, 1\}$ denote the outcome, with $y_e = 1$ indicating failure (task not solved under $\tau^2$ assertions). TRACER outputs a scalar uncertainty score at each agent decision step, $u_{e,t}$, computed from interaction-derived signals tailored to dual-control settings: (i) an observation-side signal $D_o$ capturing mismatches between expected and realized tool or world feedback, (ii) an action-side signal $D_a$ capturing deviations in action feasibility or task progress under decentralized control, and (iii) semantic and lexical similarity signals quantifying agent–user and tool-grounded misalignment.

**Evaluation metrics, baselines, and ablations.** We evaluate task failure prediction using AUROC and uncertainty-guided selective task completion using AUARC (Geifman & El-Yaniv, 2017). AUROC is computed at both the episode

*Table 2.* **AUROC / AUARC** ↑ for task failure prediction and selective task completion - $\tau^2$-bench. Best results are **bold**.

| Model | Domain | Norm. Ent. AUROC / AUARC | SelfConf AUROC / AUARC | SemEnt AUROC / AUARC | SAUP AUROC / AUARC | TRACER (ours) AUROC / AUARC |
|---|---|---|---|---|---|---|
| gemini-2.5-pro | Airline | 0.600 / 0.514 | 0.462 / 0.451 | 0.603 / 0.516 | 0.595 / 0.517 | **0.735 / 0.629** |
| | Retail | 0.553 / 0.661 | 0.440 / 0.590 | 0.556 / 0.673 | 0.525 / 0.684 | **0.673 / 0.725** |
| | Telecom | 0.649 / 0.392 | 0.507 / 0.283 | 0.651 / 0.395 | 0.639 / 0.379 | **0.691 / 0.517** |
| gemini-2.5-flash | Airline | 0.568 / 0.612 | 0.663 / 0.645 | 0.666 / 0.648 | 0.540 / 0.599 | **0.725 / 0.697** |
| | Retail | 0.468 / 0.497 | 0.530 / 0.544 | 0.533 / 0.547 | 0.428 / 0.467 | **0.707 / 0.670** |
| | Telecom | 0.559 / 0.358 | 0.423 / 0.257 | 0.621 / 0.392 | 0.673 / 0.446 | **0.809 / 0.520** |
| gpt-4.1-mini | Airline | 0.538 / 0.402 | 0.290 / 0.236 | 0.541 / 0.427 | 0.531 / 0.403 | **0.742 / 0.615** |
| | Retail | 0.617 / 0.571 | 0.577 / 0.555 | 0.620 / 0.579 | 0.609 / 0.584 | **0.689 / 0.632** |
| | Telecom | 0.548 / 0.270 | 0.683 / 0.389 | 0.686 / 0.394 | 0.534 / 0.347 | **0.765 / 0.613** |

*Table 3.* **AUROC / AUARC** ↑ for task failure prediction and selective task completion - ToolHop bench. Best results are **bold**.

| Model | Domain | Norm. Ent. AUROC / AUARC | SelfConf AUROC / AUARC | SemEnt AUROC / AUARC | SAUP AUROC / AUARC | TRACER (ours) AUROC / AUARC |
|---|---|---|---|---|---|---|
| gemini-2.5-pro | Film | 0.613 / 0.527 | 0.474 / 0.459 | 0.616 / 0.530 | 0.602 / 0.524 | **0.741 / 0.638** |
| | Genealogy | 0.573 / 0.686 | 0.467 / 0.611 | 0.579 / 0.684 | 0.550 / 0.697 | **0.692 / 0.743** |
| | Computing | 0.658 / 0.409 | 0.516 / 0.293 | 0.662 / 0.415 | 0.647 / 0.388 | **0.704 / 0.526** |
| gemini-2.5-flash | Film | 0.581 / 0.625 | 0.676 / 0.653 | 0.680 / 0.661 | 0.564 / 0.617 | **0.739 / 0.708** |
| | Genealogy | 0.482 / 0.513 | 0.545 / 0.557 | 0.549 / 0.560 | 0.443 / 0.476 | **0.721 / 0.682** |
| | Computing | 0.574 / 0.370 | 0.438 / 0.279 | 0.637 / 0.406 | 0.685 / 0.462 | **0.824 / 0.531** |
| gpt-4.1-mini | Film | 0.552 / 0.417 | 0.306 / 0.253 | 0.557 / 0.440 | 0.543 / 0.422 | **0.756 / 0.629** |
| | Genealogy | 0.631 / 0.584 | 0.590 / 0.569 | 0.634 / 0.593 | 0.626 / 0.601 | **0.705 / 0.647** |
| | Computing | 0.563 / 0.288 | 0.699 / 0.402 | 0.701 / 0.411 | 0.548 / 0.364 | **0.780 / 0.627** |
| gpt-5.4-mini | Film | 0.587 / 0.458 | 0.419 / 0.360 | 0.594 / 0.472 | 0.565 / 0.446 | **0.868 / 0.789** |
| | Genealogy | 0.646 / 0.618 | 0.603 / 0.586 | 0.651 / 0.624 | 0.635 / 0.607 | **0.846 / 0.813** |
| | Computing | 0.596 / 0.327 | 0.708 / 0.426 | 0.713 / 0.439 | 0.588 / 0.401 | **0.889 / 0.764** |

level $(s_e, y_e)$ and prefix level $(s_{e,t}, y_e)$, measuring how well uncertainty scores rank failures above successes; higher is better, with $0.5$ indicating random ranking (Hendrycks & Gimpel, 2017). We adopt AUROC due to its threshold-free evaluation and robustness under class imbalance across domains. We compare TRACER against standard uncertainty proxies, including normalized entropy, self-reported confidence, semantic entropy (Farquhar et al., 2024), and (Zhao et al., 2025). In addition, we report on ablation studies. The baselines used in our study were carefully tuned according to the recommended settings reported in their original papers and standardized to ensure a fair comparison with TRACER.

### 4.2. Failure Prediction Performance

**Task failure prediction and selective execution.** We evaluate uncertainty metrics by their ability to (i) distinguish failed tasks from successful ones and (ii) support selective task execution under uncertainty in the $\tau^2$-bench environment. Following the evaluation protocol in Sec. 3, each episode is assigned a trajectory-level uncertainty score by aggregating step-wise interaction signals (Sec. 3) over the full dialogue and labeled as success or failure using benchmark-defined assertions. We report AUROC for both episode-level $(s_e, y_e)$ and prefix-level $(s_{e,t}, y_e)$ failure prediction, where higher values indicate better ranking of failures above

successes and $0.5$ corresponds to random ordering. We additionally report AUARC, which evaluates decision quality under deferral by measuring accuracy as increasingly uncertain episodes are rejected.

Tables 2, 3 and 4 summarize both metrics across all models and domains using combined agent and user signals. TRACER achieves the highest AUROC and AUARC in every evaluated setting, outperforming all baseline uncertainty proxies. Relative to the strongest baseline, TRACER improves AUROC by 4.7%–37.1% and AUARC by 6%–55%, depending on the model and domain. Improvements are most pronounced for gpt-4.1-mini and gpt-5.4-mini, where likelihood- and confidence-based proxies fail to reliably separate easy from hard episodes. Overall, these results show that TRACER's trajectory-level uncertainty scores are both strongly discriminative and effective for selective execution, enabling reliable abstention in multi-turn, tool-using agentic settings.

**Early-warning analysis.** We evaluate whether uncertainty metrics provide actionable *early-warning signals* for impending failure. For each metric, we fix an operating threshold based on AUROC and record the first decision step at which the threshold is crossed in each failed trajectory. Detection time is normalized by total trajectory length to enable comparison across episodes of varying duration. We

*Table 4.* **AUROC / AUARC** ↑ for task failure prediction and selective task completion - ComplexFuncBench. Best results are **bold**.

| Model | Domain | Norm. Ent. AUROC / AUARC | SelfConf AUROC / AUARC | SemEnt AUROC / AUARC | SAUP AUROC / AUARC | TRACER (ours) AUROC / AUARC |
|---|---|---|---|---|---|---|
| gemini-2.5-pro | Hotels | 0.573 / 0.522 | 0.453 / 0.440 | 0.581 / 0.550 | 0.569 / 0.531 | **0.682 / 0.610** |
| | Flights | 0.610 / 0.584 | 0.485 / 0.491 | 0.622 / 0.609 | 0.600 / 0.590 | **0.715 / 0.655** |
| | Cross | 0.553 / 0.488 | 0.422 / 0.389 | 0.569 / 0.517 | 0.535 / 0.475 | **0.650 / 0.588** |
| gpt-5.4-mini | Hotels | 0.558 / 0.497 | 0.393 / 0.376 | 0.572 / 0.515 | 0.551 / 0.484 | **0.678 / 0.603** |
| | Flights | 0.595 / 0.551 | 0.455 / 0.427 | 0.616 / 0.584 | 0.592 / 0.549 | **0.712 / 0.645** |
| | Cross | 0.542 / 0.458 | 0.362 / 0.339 | 0.555 / 0.487 | 0.527 / 0.443 | **0.648 / 0.573** |
| gemini-2.5-flash | Hotels | 0.522 / 0.505 | 0.583 / 0.561 | 0.551 / 0.544 | 0.485 / 0.461 | **0.661 / 0.608** |
| | Flights | 0.559 / 0.536 | 0.628 / 0.610 | 0.597 / 0.589 | 0.515 / 0.538 | **0.693 / 0.643** |
| | Cross | 0.492 / 0.417 | 0.512 / 0.486 | 0.522 / 0.470 | 0.465 / 0.419 | **0.638 / 0.568** |
| gpt-4.1-mini | Hotels | 0.550 / 0.468 | 0.355 / 0.311 | 0.568 / 0.495 | 0.545 / 0.465 | **0.676 / 0.592** |
| | Flights | 0.593 / 0.538 | 0.413 / 0.386 | 0.613 / 0.561 | 0.588 / 0.520 | **0.729 / 0.635** |
| | Cross | 0.538 / 0.421 | 0.314 / 0.281 | 0.545 / 0.465 | 0.513 / 0.407 | **0.645 / 0.559** |

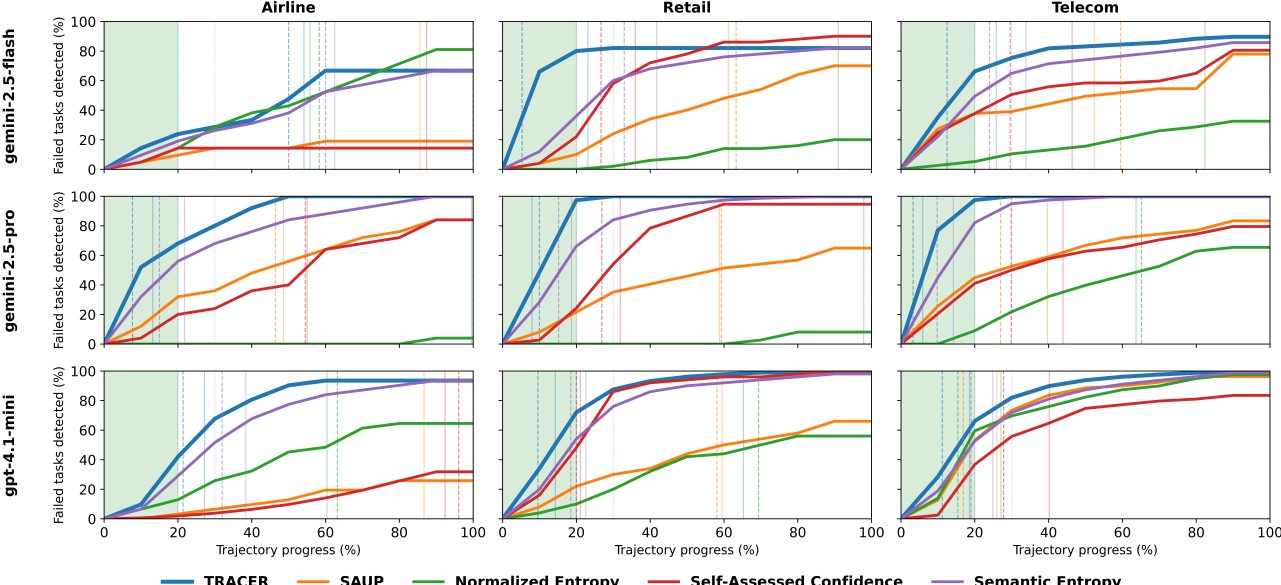

*Figure 2.* Early-warning detection curves showing the proportion of failed tasks detected by each metric as a function of trajectory progress. TRACER consistently signals failures earlier, especially within the first 20% (highlighted) of execution.

summarize performance using cumulative detection rates and mean and median normalized detection times.

Figure 2 reports cumulative failure detection as a function of trajectory progress. Across all models and domains, TRACER consistently signals failures earlier than all baselines, with the largest advantages in the first 20% of execution, highlighted in the figure, where corrective intervention remains feasible. For gemini-2.5-pro, TRACER detects failures within the first 20% of the trajectory in 68.0% (airline), 97.3% (retail), and 97.4% (telecom) of failed tasks, compared to 56.0%, 66.2%, and 82.1% for the strongest baseline (Semantic Entropy), yielding absolute gains of +12.0, +31.1, and +15.3 percentage points. Similar trends hold across other models; for example, on gpt-4.1-mini (retail), TRACER detects 72.0% of failures before 20% progress, versus 54.0% for Semantic En-

tropy and 22.0% for SAUP.

Median detection times show this pattern. TRACER typically crosses its failure threshold within 3.3%–12.5% of execution for gemini-2.5-pro and gemini-2.5-flash, whereas baseline metrics often trigger only after 25%–60% of the trajectory. These results show that TRACER is more discriminative and effective as an early-warning signal in the sparse-hazard regime that characterizes agentic breakdowns.

### 4.3. Ablation Study

**Component analysis: Agent-side vs. user-side signals.** We evaluate the contribution of agent-side and user-side uncertainty signals by selectively removing each component from TRACER. *Agent-only* retains agent normalized surprisal and coherence gaps while removing all user-derived

*Table 5.* Ablation study: AUROC (↑) for predicting task failure using **agent-only (A)** vs. **user-only (U)** signals. Per row, we bold the single best method within A and within U.

| Model | Domain | Norm. Ent. | | SelfConf | | SemEnt | | SAUP | | TRACER | |
|---|---|---|---|---|---|---|---|---|---|---|---|
| | | A | U | A | U | A | U | A | U | A | U |
| gemini-2.5-pro | Airline | 0.610 | 0.535 | 0.430 | 0.547 | 0.618 | 0.551 | 0.626 | 0.496 | **0.678** | **0.582** |
| | Retail | 0.504 | 0.573 | 0.474 | 0.428 | 0.521 | **0.576** | 0.555 | 0.524 | **0.637** | 0.489 |
| | Telecom | 0.644 | 0.555 | 0.421 | 0.555 | 0.648 | 0.571 | **0.671** | 0.593 | 0.647 | **0.622** |
| gemini-2.5-flash | Airline | 0.530 | 0.558 | 0.718 | 0.589 | **0.721** | 0.572 | 0.531 | 0.574 | 0.650 | **0.591** |
| | Retail | 0.478 | 0.453 | 0.534 | 0.506 | 0.538 | **0.510** | 0.468 | 0.465 | **0.692** | 0.4682 |
| | Telecom | 0.543 | 0.546 | 0.377 | 0.556 | 0.566 | 0.563 | 0.626 | 0.596 | **0.672** | **0.602** |
| gpt-4.1-mini | Airline | 0.582 | 0.500 | 0.421 | 0.366 | 0.585 | 0.521 | 0.590 | 0.536 | **0.602** | **0.725** |
| | Retail | 0.575 | 0.633 | 0.568 | 0.544 | 0.578 | 0.635 | 0.560 | 0.632 | **0.579** | **0.636** |
| | Telecom | 0.554 | 0.553 | 0.645 | 0.668 | 0.648 | 0.669 | 0.655 | 0.506 | **0.674** | **0.672** |

*Table 6.* TRACER aggregation variants (AUROC ↑). Best results per row are shown in **bold**.

| Model | Domain | Additive | Multiplicative | Separate | MAX |
|---|---|---|---|---|---|
| gemini-2.5-pro | Airline | 0.726 | 0.661 | 0.711 | **0.735** |
| | Retail | 0.660 | 0.582 | 0.623 | **0.673** |
| | Telecom | 0.678 | 0.681 | 0.672 | **0.691** |
| gemini-2.5-flash | Airline | 0.620 | 0.632 | 0.675 | **0.725** |
| | Retail | 0.677 | 0.500 | 0.638 | **0.707** |
| | Telecom | 0.795 | 0.785 | 0.764 | **0.809** |
| gpt-4.1-mini | Airline | 0.731 | 0.643 | 0.648 | **0.742** |
| | Retail | 0.668 | 0.627 | 0.679 | **0.689** |
| | Telecom | 0.761 | 0.689 | 0.753 | **0.765** |

signals, and *User-only* applies the converse. Table 5 reports AUROC for both. Removing either component leads to a substantial performance drop, confirming that TRACER benefits from jointly modeling both sides of the interaction. Agent-side signals carry the dominant predictive signal: removing them reduces TRACER's average gain over the strongest baseline by approximately 85%, while removing user-side signals reduces the gain by approximately 70%. User signals thus provide complementary information, capturing coordination failures and delayed misalignment not reflected in agent likelihood alone. Neither component alone recovers full performance, supporting the dual-control design.

**Aggregation variant analysis.** We compare four strategies for aggregating step-wise uncertainty signals: *additive*, *multiplicative*, *separate* weighted channels, and a *max*-composite formulation. All variants are tuned on held-out validation episodes and evaluated using AUROC on test trajectories. As shown in Table 6, the MAX variant achieves the highest AUROC across all model–domain pairs. This result reflects the failure structure of agentic interaction, where breakdowns are driven by sparse, decisive uncertainty spikes rather than sustained moderate uncertainty. Max-based aggregation preserves these dominant signals, whereas additive or multiplicative coupling dilutes them, empirically supporting TRACER's tail-focused design.

## 5. Conclusion

We introduced TRACER, a trajectory-level uncertainty metric for multi-turn, tool-using agentic interaction. TRACER models failure as arising from sparse critical episodes rather than uniformly elevated token uncertainty, combining content-aware surprisal with situational-awareness signals and tail-focused aggregation. Across $\tau^2$-bench domains and models, TRACER consistently improves failure separability, selective execution, and early-warning detection relative to likelihood- and confidence-based proxies. Ablations confirm the necessity of dual-control (agent and user) signals and the effectiveness of MAX-based aggregation for capturing decisive breakdowns. Together, these results position TRACER as a principled and practical uncertainty measure for real-world agentic systems, enabling earlier intervention and more reliable abstention in complex interactive settings.

## Acknowledgement

This work was supported by COGNISENSE, part of the JUMP 2.0 program sponsored by DARPA and NSF Award 2329096.

## Impact Statement

This work introduces TRACER, a trajectory-level uncertainty metric for multi-turn, tool-using agentic systems. By focusing on sparse but decisive failure episodes rather than average token-level uncertainty, TRACER enables earlier and more reliable detection of breakdowns in interactive settings, supporting safer deployment in applications such as customer support, troubleshooting, and decision assistance. TRACER is a diagnostic and monitoring method rather than a decision-making system. It does not generate content or actions, but provides an uncertainty signal that can trigger human oversight, fallback strategies, or verification. The method relies only on signals already produced during agent interaction and does not require additional user data. Potential negative impacts include misuse of uncertainty scores as hard decision thresholds, leading to over-abstention or

reduced utility. We mitigate this risk through selective prediction and early-warning evaluation and by providing theoretical guarantees that clarify TRACER's scope. Overall, this work contributes to more reliable and safety-aware agentic systems by aligning uncertainty estimation with how failures arise in practice.

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

# A. Formal Theorems: Calibration, Coherence, and Failure-Risk Control

This appendix provides theoretical justification that TRACER is a mathematically principled metric for estimating uncertainty in multi-turn, task-oriented, dual-control dialogue. We establish (i) an information-theoretic interpretation of the content-aware surprisal $U_t$, (ii) coherence of the tail-aggregation functional, (iii) stability and robustness guarantees, and (iv) conditions under which TRACER upper-bounds trajectory-level breakdown risk.

## A.1. Probabilistic Setup

Let $(\Omega, \mathcal{F}, \mathbb{P})$ be a probability space supporting the trajectory random variable $\mathcal{T}$ induced by the environment and the two stochastic policies $(\pi_A, \pi_U)$. Let $\{\mathcal{F}_t\}_{t=0}^N$ denote the filtration generated by the partial history up to step $t$:

$$\mathcal{F}_t \triangleq \sigma\big(\{(a_s, x_s, o_s)\}_{s=1}^t\big), \qquad \mathcal{F}_0 \triangleq \{\emptyset, \Omega\}. \tag{25}$$

We model *uncertainty* as a possibly latent stepwise random variable $C_t \in [0, 1]$ measurable with respect to $\mathcal{F}_t$, where larger values indicate greater uncertainty, such as looping, incoherent tool invocation, or user-agent miscoordination at step $t$. The overall *breakdown* event is

$$B \triangleq \left\{ \max_{t \in \{1, \ldots, N\}} C_t > 0 \right\}. \tag{26}$$

This formalizes that a single critical episode can derail an otherwise successful task-oriented conversation.

## A.2. Information-Theoretic Interpretation of Content-Aware Surprisal

For a generative step $t$, let $Q_{t,j}(\cdot) \triangleq \mathbb{P}(W_{t,j} = \cdot \mid W_{t,<j}, C_t)$ denote the true conditional token distribution, and let $P_{t,j}(\cdot)$ denote the model distribution used to compute $p_t(\cdot)$. Define the content-filtered cross-entropy

$$H_t^{\mathrm{cont}}(Q, P) \triangleq \frac{1}{|\mathcal{I}_t|} \sum_{j \in \mathcal{I}_t} \mathbb{E}_{W \sim Q_{t,j}}\big[-\log P_{t,j}(W)\big], \tag{27}$$

with a benign convention when $|\mathcal{I}_t| = 0$.

**Theorem A.1** (Decomposition to uncertainty and mismatch). *Assume $|\mathcal{I}_t| > 0$ and $Q_{t,j} \ll P_{t,j}$ for all $j \in \mathcal{I}_t$:*

$$H_t^{\mathrm{cont}}(Q, P) = \underbrace{\frac{1}{|\mathcal{I}_t|} \sum_{j \in \mathcal{I}_t} H(Q_{t,j})}_{\textit{intrinsic (aleatoric) content uncertainty}}$$

$$+ \underbrace{\frac{1}{|\mathcal{I}_t|} \sum_{j \in \mathcal{I}_t} \mathrm{KL}(Q_{t,j} \,\|\, P_{t,j})}_{\textit{epistemic mismatch on content tokens}}. \tag{28}$$

*Here $H(\cdot)$ denotes Shannon entropy and $\mathrm{KL}(\cdot\|\cdot)$ denotes Kullback-Leibler divergence. Moreover, the sample statistic $U_t$ is an unbiased estimator of $H_t^{\mathrm{cont}}(Q, P)$ conditional on $(W_{t,<j}, C_t)$.*

*Proof.* For each $j \in \mathcal{I}_t$,

$$\mathbb{E}_{W \sim Q_{t,j}}[-\log P_{t,j}(W)] = \mathbb{E}_{W \sim Q_{t,j}}[-\log Q_{t,j}(W)]$$
$$+ \mathbb{E}_{W \sim Q_{t,j}}\left[\log \frac{Q_{t,j}(W)}{P_{t,j}(W)}\right]$$
$$= H(Q_{t,j}) + \mathrm{KL}(Q_{t,j} \,\|\, P_{t,j}).$$

Averaging over $j \in \mathcal{I}_t$ yields (28). Unbiasedness of $U_t$ follows from its definition as the empirical mean of $-\log P_{t,j}(W_{t,j})$ over $j \in \mathcal{I}_t$. $\square$

**Implication.** Theorem A.1 formally justifies $U_t$ as a content-focused uncertainty proxy: it increases either when the underlying next-token distribution is intrinsically high-entropy or when the model distribution mismatches the true distribution, and it is not diluted by predictable structural tokens.

## A.3. Coherence of the TRACER Tail Functional

Define $\rho_{k,w} : \mathbb{R}^N \to \mathbb{R}$ by

$$\rho_{k,w}(\mathbf{r}) \triangleq (1-w)\,\mathrm{TM}_k(\mathbf{r}) + w\,\|\mathbf{r}\|_\infty. \tag{29}$$

**Theorem A.2** (Coherence of $\rho_{k,w}$). *For any $k \in (0,1]$ and $w \in [0,1]$, $\rho_{k,w}$ is a coherent risk functional on $\mathbb{R}^N$: it satisfies monotonicity, translation invariance, positive homogeneity, and subadditivity.*

*Proof.* Both $\mathrm{TM}_k$ (empirical CVaR / tail mean) and $\|\cdot\|_\infty$ are monotone, translation invariant, and positively homogeneous; $\|\cdot\|_\infty$ is a norm hence subadditive, and $\mathrm{TM}_k$ is a coherent tail-risk functional (subadditive). A convex combination preserving these properties yields the claim. $\square$

## A.4. Stability to Local Perturbations

Because $U_t$ is computed from finite-length generations and $D_a, D_o$ are embedding-based distances, the per-step components are noisy. The MAX composition is itself stable.

**Lemma A.3** (Max-aggregation is nonexpansive). *Let $(u_1, \ldots, u_m)$ and $(v_1, \ldots, v_m)$ be vectors in $\mathbb{R}^m$. Then*

$$\left| \max_{i \le m} u_i - \max_{i \le m} v_i \right| \le \max_{i \le m} |u_i - v_i|.$$

*In particular, if each component is perturbed by at most $\varepsilon$, then $|r_t - r_t'| \le \varepsilon$.*

*Proof.* Let $i^\star \in \arg\max_i u_i$. Then

$$\max_i u_i - \max_i v_i = u_{i^\star} - \max_i v_i \le u_{i^\star} - v_{i^\star} \le \max_i |u_i - v_i|.$$

Swap $u$ and $v$ to bound the absolute value. $\square$

**Theorem A.4** ($\ell_\infty$-Lipschitz stability). *Let $\mathbf{r}, \mathbf{s} \in \mathbb{R}^N$. Then for any $k \in (0,1]$ and $w \in [0,1]$,*

$$\left| \rho_{k,w}(\mathbf{r}) - \rho_{k,w}(\mathbf{s}) \right| \le \|\mathbf{r} - \mathbf{s}\|_\infty. \tag{30}$$

*Proof.* Since order statistics are 1-Lipschitz in $\|\cdot\|_\infty$, we have $|r_{(i)} - s_{(i)}| \le \|\mathbf{r} - \mathbf{s}\|_\infty$ for all $i$. Therefore,

$$|\mathrm{TM}_k(\mathbf{r}) - \mathrm{TM}_k(\mathbf{s})| \le \|\mathbf{r} - \mathbf{s}\|_\infty.$$

Moreover,

$$\left| \|\mathbf{r}\|_\infty - \|\mathbf{s}\|_\infty \right| \le \|\mathbf{r} - \mathbf{s}\|_\infty.$$

Combining both terms yields (30). $\square$

## A.5. TRACER Upper-Bounds Breakdown Risk under Sparse Hazards

Define the conditional hazard

$$\lambda_t \triangleq \mathbb{P}(C_t > 0 \mid \mathcal{F}_{t-1}) \in [0,1]. \tag{31}$$

**Lemma A.5** (Union bound for breakdown probability). *With $B$ defined in (26),*

$$\mathbb{P}(B) \le \mathbb{E}\left[ \sum_{t=1}^N \lambda_t \right].$$

*Proof.* $\mathbf{1}_B \le \sum_{t=1}^N \mathbf{1}\{C_t > 0\}$. Taking expectations and applying the tower property yields the claim. $\square$

**Assumption A.6** (Risk-dominates hazard (MAX form)). *There exists $c > 0$ such that for all $t$,*

$$\lambda_t \le \min\{1,\, c\,r_t\}.$$

**Assumption A.7** (Tail-sparsity of critical episodes). *For $K$ defined in (21), there exists $\eta \geq 0$ such that*

$$\sum_{i=K+1}^{N} r_{(i)} \leq \eta.$$

**Theorem A.8** (Breakdown risk control by tail mean). *Under Assumptions A.6–A.7,*

$$\mathbb{P}(B) \leq c\left(K\,\mathbb{E}[\mathrm{TM}_k(\mathbf{r}(\mathcal{T}))] + \eta\right).$$

*Moreover, since* $\mathrm{TRACER}_{\boldsymbol{\theta}}(\mathcal{T}) \geq (1-w)\mathrm{TM}_k(\mathbf{r}(\mathcal{T}))$, *we also have*

$$\mathbb{P}(B) \leq \frac{cK}{1-w}\,\mathbb{E}[\mathrm{TRACER}_{\boldsymbol{\theta}}(\mathcal{T})] + c\eta, \qquad w < 1.$$

*Proof.* By Lemma A.5 and Assumption A.6,

$$\mathbb{P}(B) \leq c\,\mathbb{E}\left[\sum_{i=1}^{N} r_{(i)}\right] = c\,\mathbb{E}\left[\sum_{i=1}^{K} r_{(i)} + \sum_{i=K+1}^{N} r_{(i)}\right].$$

The first term equals $cK\,\mathbb{E}[\mathrm{TM}_k(\mathbf{r})]$ and the second is bounded by $c\eta$. $\qquad\square$

## A.6. Dual-Control Decomposition and Actor-Wise Attribution

Define actor-wise step risks

$$r_t^{\mathrm{A}} \triangleq \mathbf{1}\{a_t = \text{Agent}\}\max\{U_t,\ \alpha D_a(t),\ \beta D_o^{\mathrm{A}}(t)\}, \tag{32}$$

$$r_t^{\mathrm{U}} \triangleq \mathbf{1}\{a_t = \text{User}\}\max\{U_t,\ \gamma D_o^{\mathrm{U}}(t)\}. \tag{33}$$

Then $r_t = r_t^{\mathrm{A}} + r_t^{\mathrm{U}}$.

**Theorem A.9** (Subadditive actor-wise attribution).

$$\mathrm{TRACER}_{\boldsymbol{\theta}}(\mathcal{T}) \leq \mathrm{TRACER}_{\boldsymbol{\theta}}(\mathcal{T}^{\mathrm{A}}) + \mathrm{TRACER}_{\boldsymbol{\theta}}(\mathcal{T}^{\mathrm{U}}).$$

*Proof.* Since $\mathbf{r} = \mathbf{r}^{\mathrm{A}} + \mathbf{r}^{\mathrm{U}}$ componentwise, subadditivity of $\rho_{k,w}$ yields the result. $\qquad\square$

## A.7. Parameter Identification via Separation Objectives

TRACER depends on $\boldsymbol{\theta} \triangleq (\alpha, \beta, \gamma, k, w)$. Given labeled trajectories $\mathcal{D} = \{(\mathcal{T}^{(i)}, y^{(i)})\}_{i=1}^{M}$, we define $s_{\boldsymbol{\theta}}^{(i)} = \mathrm{TRACER}_{\boldsymbol{\theta}}(\mathcal{T}^{(i)})$ and minimize the pairwise logistic loss

$$\mathcal{L}(\boldsymbol{\theta}) = \frac{1}{|\mathcal{F}||\mathcal{S}|}\sum_{i\in\mathcal{F}}\sum_{j\in\mathcal{S}}\log\left(1 + \exp\left(-\tfrac{1}{\tau}(s_{\boldsymbol{\theta}}^{(i)} - s_{\boldsymbol{\theta}}^{(j)})\right)\right).$$

We adopt a discrete coarse-to-fine grid search over $(\alpha, \beta, \gamma, k)$ followed by calibration of $w$. Empirically, MAX aggregation often emphasizes $D_o^{\mathrm{A}}$ in failed trajectories; tuned configurations therefore assign larger $\beta$ while suppressing weaker predictors to reduce noise.

## A.8. Computational Complexity

Let $L_t$ denote the token length at step $t$. Computing content-aware surprisal costs $\mathcal{O}(\sum_t L_t)$. Situational metrics require embedding computation with cost

$$\mathcal{O}\left(\sum_{t=1}^{N} c_\varphi(z(x_t)) + \sum_{t=1}^{N} c_\varphi(z(o_t))\right).$$

MAX step composition costs $\mathcal{O}(N)$, and tail aggregation requires $\mathcal{O}(N\log N)$ time. Overall complexity is

$$\mathcal{O}\left(\sum_{t=1}^{N} L_t + \sum_{t=1}^{N} c_\varphi(z(x_t)) + \sum_{t=1}^{N} c_\varphi(z(o_t)) + N\log N\right).$$

