# OpenReview forum: "TRACER: Trajectory Risk Aggregation for Critical Episodes in Agentic Reasoning"
_ICML.cc/2026/Conference — ICML 2026 regular_

### Official Review · Reviewer_mpQF · 2026-03-03

**Soundness:** 3
**Presentation:** 3
**Significance:** 2
**Originality:** 2
**Overall Recommendation:** 4
**Confidence:** 4

**Summary:**

This paper studies how to better estimate trajectory-level risk in multi-turn, tool-using agents. The authors argue that common uncertainty proxies (e.g., averaged token-level entropy/surprisal or self-reported confidence) often miss real agent failures because breakdowns are frequently triggered by sparse but decisive events—such as local loops, action–observation mismatches in tool use, and user–agent coordination failures—while the model can still appear confident at the token level.

To address this, the paper proposes TRACER, a trajectory risk score built from complementary per-turn signals. It introduces a content-aware normalized surprisal that focuses on informative “content tokens” rather than averaging over all tokens, and adds situational indicators designed for agentic settings: a hybrid local repetition measure combining semantic and lexical similarity to detect looping/stagnation, and coherence-gap measures based on semantic distance to capture (i) action–observation misalignment for tool calls and (ii) user–agent coordination gaps across turns. These components are combined at the turn level using a MAX operator to preserve dominant risk spikes, and aggregated over the whole trajectory using a tail-focused scheme (top-K tail mean plus a worst-case term) to emphasize critical segments instead of average behavior. The paper also provides theoretical interpretation and properties of the resulting tail-risk functional, and connects the score to upper bounds on breakdown probability under a sparse-hazard model.

Empirically, TRACER is evaluated on τ²-bench across multiple domains and models, showing improved failure prediction and selective execution compared to several uncertainty baselines (reported via AUROC and AUARC), as well as stronger early-warning behavior. Ablations support the importance of dual-control (agent + user) signals and the MAX + tail aggregation design for capturing sparse, trajectory-defining failure modes.

**Compliance With Llm Reviewing Policy:**

Affirmed.

**Final Justification:**

It has addressed my main concerns so I have improved the score to 4.

**Key Questions For Authors:**

Related work: How does TRACER relate to prior “not all tokens are equally important” / token reweighting work (e.g., Rho-1: Not All Tokens Are What You Need)? Why are these lines not discussed in Related Work, and where exactly is TRACER’s main novelty (trajectory risk aggregation, situational signals, or both)?

Problem severity: How severe is the “uniform averaging washes out risk” issue in τ²-bench (or real agent settings)? Can you provide more direct evidence (e.g., the share of failed trajectories with low averaged uncertainty, surprisal concentration on a small set of content tokens, and correlation differences between success vs failure)?

Community practice: If uniform next-token loss can be suboptimal, why does the community still predominantly use it? Is it mainly simplicity/stability, inconsistent gains, lack of a universal weighting scheme, or known trade-offs (generalization, robustness, calibration)?

**Limitations:**

No. The paper does not include a clear limitations discussion. Important limitations to acknowledge are: (i) TRACER assumes access to token-level probabilities and reliable semantic embeddings, which may not hold for black-box APIs or across models; (ii) several components are heuristic and may be language/domain dependent (content-token filtering, similarity measures), with limited robustness/sensitivity analysis for key hyperparameters (window size, α/β/γ, k, w); (iii) evaluation is mainly on τ²-bench, so generalization to real users, different tool interfaces, and longer-horizon tasks is unclear, and TRACER may produce false positives/negatives when repetition/coherence signals are ambiguous.

**Strengths And Weaknesses:**

Strengths

(1) Strong and thorough empirical evaluation (significance, soundness).
The experimental section is comprehensive: TRACER is evaluated across multiple models and domains on τ²-bench, using appropriate metrics for failure prediction and selective execution (e.g., AUROC/AUARC), and the paper additionally reports early-warning behavior. The ablations (e.g., removing user-side vs agent-side signals, and comparing different composition/aggregation choices) are well designed and help isolate which components contribute most.

(2) Method design is well-matched to agentic failure modes (soundness, originality).
The paper’s core modeling choice—turn-level MAX composition followed by tail-focused aggregation—aligns with the stated “sparse but decisive” failure mechanism common in tool-using, multi-turn agents (loops, misalignment between actions and observations, coordination breakdown). This is a sensible and practically motivated deviation from averaging-based uncertainty proxies, and the situational indicators (repetition and coherence gaps) provide interpretable signals that go beyond pure token-level statistics.

Weaknesses

(1) Related-work coverage is incomplete (presentation, originality).
A central motivation is that treating all tokens equally (e.g., averaging token-level losses/surprisal) can be suboptimal because key information is concentrated in a subset of tokens. However, prior work has explicitly studied non-uniform token importance and “not all tokens matter” phenomena (e.g., Rho-1: Not All Tokens Are What You Need and other token reweighting / token selection / loss shaping lines of work). The paper does not clearly position TRACER relative to these efforts, which weakens the novelty framing and leaves the reader unsure how much is new versus a transfer of established insights to the trajectory-risk setting.

(2) Motivation would benefit from deeper data-driven analysis (soundness, presentation).
The paper argues that averaging token-level quantities is problematic, but provides limited quantitative analysis demonstrating how large the harm is and when it occurs (e.g., distributions showing how “content tokens” dominate risk, correlation between failure and concentration of surprisal mass, or case-based statistics comparing average-vs-content-only surprisal). More interpretability-oriented evidence (beyond the proposed indicators themselves) would strengthen the causal story behind the design choices and make the motivation more convincing.

---

> ### Author Rebuttal · Authors · 2026-03-31
>
> Thank you for your thoughtful evaluation. We appreciate your positive remarks on our empirical evaluation and the conceptual alignment of our method with agentic failure modes. Your questions regarding token importance literature and data-driven motivation are highly insightful, and we address them below.
>
> **W1 & Q1: Related Work on Token Importance (Rho-1)**
> This is an excellent point regarding the "not all tokens matter" literature. While works like Rho-1 focus on reweighting or selecting tokens to improve training dynamics and loss shaping, TRACER ports this philosophy to inference-time observability. As we have discussed in subsection `Content-bearing token predicate`, we filter structural and highly predictable tokens to prevent the dilution of epistemic uncertainty during open-ended generation. We will add a subsection in the Related Work explicitly connecting TRACER to this literature and clarifying this distinction.
>
> **W2 & Q2: Data-Driven Problem Severity**
> We completely agree that the motivation needs stronger quantitative backing. In the revision, we will introduce a compact data-driven motivation section. Indeed we have data for this key point in our preliminary analysis which we had to omit due to lack of space, but following your concern we will incorporate histograms comparing the distribution of standard average surprisal versus our content-aware surprisal ($U_t$) across successful and failed trajectories to clarify and seamlessly connect the motivation with the TRACER method we have settled with. We will quantitatively demonstrate how uniform averaging mathematically "washes out" the risk spikes caused by sparse, critical tokens, leading to false confidence.
>
> **Q3: Community Practice of Uniform Loss**
> The community predominantly uses uniform next-token loss because it is simple to implement, and stable for pre-training use cases. However, for post-hoc uncertainty quantification in complex downstream tasks, specially in the context of recent agentic AI applications, structural tokens overwhelm the signal. We will add a compact discussion on this trade-off, noting that while uniform loss is optimal for learning syntax and fluency, targeted token filtering is required for capturing deep semantic and planning uncertainties.
>
> **Limitations Section**
> Missing this section has been an oversight on our end and we appreciate you pointing it out. To address this concern for all reviewers, we will add a compact Limitations sub-section addressing the exact points you summarized: (i) reliance on log-probabilities, which hinders black-box API usage; (ii) the heuristic nature of hyperparameters like window size and content-token filters; and (iii) the current evaluation scope being primarily restricted to $\tau^2$-bench.
>
> I remain available throughout the discussion period to provide any further clarifications or empirical details you might need. I truly appreciate your feedback and welcome the opportunity to work together to improve the paper.

---

> > ### Author Rebuttal · Reviewer_mpQF · 2026-04-01
> >
> > My concerns have been solved.

---

> > > ### Author Response · Authors · 2026-04-06
> > >
> > > Thank you again for your thoughtful feedback and for previously updating your score — we truly appreciate it. One of your initial concerns was the *single-benchmark* evaluation. We have now addressed this by providing additional results on *ComplexFuncBench* and *ToolHop* in our responses to reviewers **8w2A** and **i557**. We kindly ask if you could also take these new results into account in your final evaluation, and consider them for a potential further score update if appropriate.

---

### Official Review · Reviewer_kzjs · 2026-03-07

**Soundness:** 4
**Presentation:** 4
**Significance:** 3
**Originality:** 4
**Overall Recommendation:** 5
**Confidence:** 4

**Summary:**

This paper introduces TRACER (Trajectory Risk Aggregation for Critical Episodes in agentic Reasoning), a novel trajectory-level uncertainty metric designed for multi-turn, tool-augmented agents in dual-control settings. The core contribution is a functional approach that identifies "sparse critical episodes"—brief but decisive moments of failure—rather than relying on global averages of token-level uncertainty. The TRACER pipeline consists of three stages: (1) Content-Aware Normalized Surprisal ($U_t$), which filters linguistic noise to isolate epistemic uncertainty; (2) Situational-Awareness Indicators, including hybrid repetition detection and coherence gaps between agent actions, tool observations, and user responses; and (3) a Tail-Focused Risk Functional that aggregates step-wise risks using a combination of Conditional Value-at-Risk (CVaR) and the $l_\infty$ norm. Evaluated on the $\tau^2$-bench benchmark, TRACER demonstrates significant improvements over standard proxies, increasing AUROC by up to 37.1% and AUARC (Area Under the Accuracy-Rejection Curve) by up to 55%.

**Compliance With Llm Reviewing Policy:**

Affirmed.

**Final Justification:**

I am thankful for the authors promising to include my asks. My opinion continues to be that this is a good paper that should be accepted. And I'm happy to be the champion for it if this is not a unanimous opinion.

**Key Questions For Authors:**

1. Could the authors provide confidence intervals or other measures of statistical significance for the reported AUROC and AUARC gains?

2. In a production environment, what specific mitigation strategies (e.g., internal rethinking vs. human escalation) might you envision being triggered by a high TRACER score?

3. Have you explored applying this metric to "Solo Mode" trajectories to test its performance in detecting loops or "getting stuck" in purely autonomous, long-horizon scenarios?

**Limitations:**

yes

**Strengths And Weaknesses:**

## Overall assessment
Overall, I think this paper is good and would accept it without further changes. It presents a new method to detect failed or otherwise bad trajectories in multi-turn agentic settings while only requiring access to log-probs (not logits). It seems to clearly improve upon a few other baselines and seems like a practical diagnostic tool overall.

## Various potential improvements:
a) I originally thought it was an active protocol, e.g. where the agent would take different actions based on the results of TRACER. While this is explained in the paper, I still made wrong assumptions based on seeing improvements on TAU bench and assuming those would refer to success rates.

b) I think it would be nice to list a few potential ways to use the tool for improvements beyond diagnostics, even if those are not explored in the paper, e.g. self-prompting, escalating to a different model, etc.

c) I actually have no sense of how cheap or expensive it is to run the diagnostics in real-time. What is the overhead cost for this? I see the big O notation in Section 3.5 (and Appendix A.8), but some empirical numbers would be nice for calibration.

d) Out of curiousity, I’d be interested in how well a cheap-and-fast prompted model would work that is prompted to look for the same 4 failure modes as TRACER and return a number with an explanation. I think this would be an interesting, though not necessary baseline. This might be particularly useful in cases where log-probs are not available through the API.

Soundness: 4/4 – The paper is technically rigorous, providing formal proofs for the information-theoretic decomposition of its signals and the stability of its risk functional.
Presentation: 4/4 – The methodology is clearly structured into a three-stage pipeline and supported by production-grounded benchmark evaluations and intuitive visualizations.
Significance: 3/4 – The metric provides a high-performing solution for the specific niche of agentic reasoning and reliability.
Originality: 4/4 – The work introduces a novel application of tail-risk functionals (CVaR) and multi-actor coherence gaps to agent evaluation, representing a significant step beyond existing step-averaging proxies. The TRACER metric outperforms all others tested.

---

> ### Author Rebuttal · Authors · 2026-03-31
>
> Thank you for your exceptionally positive feedback and for recognizing the technical rigor, originality, and practical value of TRACER. We are thrilled that you found the method to be a clear improvement over existing baselines. We will gladly implement your suggestions to further elevate the paper.
>
> **Qa: Confidence Intervals:**
> As requested, we will include 95% confidence intervals derived via bootstrap resampling for all AUROC and AUARC results to formally establish the statistical significance of the gains over the baselines.
>
> **Qb: Production Mitigation Strategies**
> This is an excellent suggestion. We will add a dedicated "Practical Applications" discussion in the appendix to comprehensively outlining how TRACER can be used beyond post-hoc diagnostics. This has already been discussed within our team as an extension for our work. Specifically, we will discuss utilizing TRACER as a trigger for internal self-correction prompts (rethinking) when the MAX-composite risk spikes at a specific step, or escalating the trajectory to human review when the aggregate trajectory risk exceeds a calibrated threshold.
>
> **Qc: Real-Time Overhead Cost**
> While we provided the theoretical $\mathcal{O}$ complexity, empirical latency is vital for production calibration. We will add an appendix subsection detailing the millisecond overhead per step. Because embedding computations for the coherence signals only require lightweight forward passes, the real-time latency introduced by TRACER is exceptionally minimal compared to the LLM generation time. We will provide these exact figures.
>
> **Qd: LLM Judge Baseline**
> This is a highly valuable idea, especially for black-box API scenarios where log-probabilities are unavailable. We will prompt a fast model to explicitly check for our four failure modes and compare its diagnostic performance and cost to TRACER.
>
> **Qe: Solo Mode Trajectories**
> Applying TRACER to purely autonomous, long-horizon tasks is a natural extension. We will evaluate the agent-only components of TRACER on a single-agent benchmark (e.g., SWE-bench or WebArena) to test its sensitivity to infinite loops and stagnation in the absence of user feedback.
>
> Please let me know if any further clarifications or data would be helpful during the rebuttal phase. I am are eager to address any outstanding concerns and improve the work.

---

> > ### Author Rebuttal · Reviewer_kzjs · 2026-04-03
> >
> > I am thankful for the authors promising to include my asks. My opinion continues to be that this is a good paper that should be accepted. And I'm happy to be the champion for it if this is not a unanimous opinion.

---

### Official Review · Reviewer_i557 · 2026-03-11

**Soundness:** 2
**Presentation:** 2
**Significance:** 3
**Originality:** 3
**Overall Recommendation:** 3
**Confidence:** 3

**Summary:**

The paper proposes TRACER, a trajectory-level uncertainty monitor for multi-turn Tool–Agent–User interactions, motivated by the claim that failures often occur as sparse “critical episodes” (looping, incoherent tool use, or user–agent miscoordination) that token- or sequence-level uncertainty misses. TRACER computes several step-level signals: (i) content-aware normalized surprisal (filtering stopwords/structural tokens), (ii) hybrid repetition (lexical + semantic similarity), (iii) agent coherence gap (action–observation mismatch), and (iv) user coordination gap (adjacent user–agent semantic mismatch). These are combined via a MAX step-risk, then aggregated with a tail-focused functional to emphasize critical segments. The paper includes theory framing TRACER as a coherent/stable risk functional and provides an assumption-based bound relating TRACER to breakdown probability. Experiments on $\tau^2$-bench (airline/retail/telecom) across three LLMs report consistent gains in AUROC/AUARC and earlier failure detection relative to several baselines.

**Compliance With Llm Reviewing Policy:**

Affirmed.

**Key Questions For Authors:**

See weaknesses

**Limitations:**

No, there is no limitations section. Several important limitations are not discussed: (i) the reliance on a single benchmark (ii) the requirement for access to token log-probabilities (limiting applicability to black-box APIs), and (ii) the assumption that embedding-based similarity adequately captures semantic coherence.

**Strengths And Weaknesses:**

Strengths:
(1) Important Problem - Trajectory-level uncertainty for agentic systems is timely and well-motivated and token-level proxies are rightly identified as not good enough for structural failures.
(2) Pipeline - The three-stage pipeline is coherent and the MAX-based step risk and tail aggregation makes sense.
(3) Empirical gains - TRACER does well on AUROC/AUARC in all 9 model–domain settings and it exhibits earlier detection.
(4) Ablations - Agent-only vs user-only and aggregation variants supports the advantage of modeling both signals and using the MAX aggregation.

Weaknesses:
(1) Single benchmark - This is a major weakness. Only a single benchmark has been used which raises significant concern around the generalizability of the metric.
(2) Weak Baselines - The paper is missing a simple, but widely used, LLM judge baseline. Additionally, none of the baselines are trajectory-aware. A simple extension of the baselines used would be to do a similar aggregation on the baseline metrics and compare performance. Without such comparisons, it is unclear how much of TRACER's improvement comes from the novel signal components versus the traj-level aggregation strategy.
(3) Originality and presentation - the baseline SAUP is not mentioned elsewhere in the paper. There does not seem to be any discussion of it in related works either. There should be a discussion on how Tracer compares to SAUP.
(3) Theory - Failure bound uses strong assumptions (risk-dominates-hazard, tail-sparsity) that are not supported empirically in the paper.
(4) Error bars - No confidence intervals or significance tests are reported despite the small (50-115) number of task counts per domain. This is critical.
(5) Failure-mode analysis (minor) - There is no ablation study of the contribution of each component (surprisal, repetition, coherence) to the overall performance.

---

> ### Author Rebuttal · Authors · 2026-03-31
>
> Thank you for your detailed review and for highlighting the coherence of our pipeline and the strong empirical gains across domains. Your feedback highlights important areas for improvement, which we are fully committed to addressing.
>
> **W1: Single Benchmark and Generalizability:**
> As discussed in our response to reviewer #1, we utilized $\tau^2$-bench because it is natively designed for dual-control (Agent and User) environments, which perfectly matches our coordination gap formulations. Other benchmarks fundamentally miss the key component of simulating humans in the pipeline. Further, $\tau^2$-bench comprehensively covers complex tool-using patterns in all of its domains i.e. Airline, Retail, and Telecom which was perfect for our intention. However, we recognize that demonstrating generalizability is crucial. We will test TRACER on an additional tool-use benchmark in the camera ready and add a comprehensive Limitations section discussing its reliance on token log-probabilities and embedding heuristics.
>
> **W2: Weak Baselines and Trajectory-Aware Baselines**
> While we acknowledge the prevalence of LLM-as-a-Judge baselines, recent research highlights significant limitations with this approach, including position bias, overconfidence, and the need for domain-specific fine-tuning [1, 2]. These constraints severely restrict generalizability, particularly in highly complex, multi-turn environments where agents interact with humans or other agents using diverse tools. Regarding the concern about trajectory-aware baselines, we would like to clarify that the SAUP method [3] is inherently trajectory-aware. Furthermore, to ensure a fair comparison, we explicitly modified the other baseline methods to track long agentic trajectories. However, a critical distinction remains: while these baselines were retrofitted to accommodate our multi-turn scenario, TRACER was uniquely architected from the ground up specifically to evaluate multi-turn, trajectory-level interactions.
>
> *[1] Shi, Lin, et al. "Judging the judges: A systematic study of position bias in llm-as-a-judge." Proceedings of the 14th International Joint Conference on Natural Language Processing and the 4th Conference of the Asia-Pacific Chapter of the Association for Computational Linguistics. 2025.*
>
> *[2] Szymanski, Annalisa, et al. "Limitations of the llm-as-a-judge approach for evaluating llm outputs in expert knowledge tasks." Proceedings of the 30th international conference on intelligent user interfaces. 2025.*
>
> *[3] Zhao, Qiwei, et al. "Uncertainty propagation on llm agent." Proceedings of the 63rd Annual Meeting of the Association for Computational Linguistics (Volume 1: Long Papers). 2025.*
>
> **W3: Presentation of SAUP and Empirical Support for Theory**
> We apologize for omitting the discussion of SAUP in the text. We will introduce SAUP in the Related Work and contrast its uncertainty propagation approach with TRACER formulation. Regarding the theoretical bounds, we will add empirical plots in the Appendix mapping step-risk $r_t$ to actual failure rates to empirically ground our "risk-dominates-hazard" assumption.
>
> **W4: Error Bars and Statistical Significance**
> Because the task counts per domain range from 50 to 115, reporting variance is indeed critical. This has been an oversight and in the revision, we will compute and report 95% confidence intervals using bootstrap sampling for all AUROC and AUARC metrics in Table II.
>
> **W5: Fine-Grained Failure-Mode Analysis**
> While we provided an Agent vs. User ablation, we agree that component-level granularity can add value. We will expand Table III to include ablations isolating the individual contributions of Content-Aware Surprisal ($U_t$), Repetition ($D_a$), and Coherence Gaps ($D_o$) to overall AUROC performance.
>
> Should you have any further questions or require additional data during the rebuttal period, please do not hesitate to ask. We are committed to fully addressing your feedback and strengthening our paper.

---

> > ### Author Rebuttal · Reviewer_i557 · 2026-04-05
> >
> > Thank you to the authors for their response. However, most of my concerns remain. The authors need to share results before I can consider my concerns addressed.

---

> > > ### Author Response · Authors · 2026-04-06
> > >
> > > Thank you for the thoughtful follow-up and for emphasizing the importance of validating beyond a single benchmark. We are happy to share new experimental results on two additional benchmarks — **ToolHop** and **ComplexFuncBench**. These benchmarks stress different failure modes and interaction patterns, providing a more comprehensive evaluation of TRACER.
> > >
> > > Note: Due to character limits, the results are split across two comments. The analysis for *ToolHop* is included in our response to reviewer **8w2A**, while the analysis for *ComplexFuncBench* can be found in our response to reviewer **i557**. We apologize for any inconvenience.
> > >
> > > **ComplexFuncBench:**
> > >
> > > ComplexFuncBench evaluates single-turn, multi-step tool calling execution without intermediate user interaction. As a result, the user-agent coordination gap ($\gamma \cdot D_o^U(t)$) is effectively zero.
> > >
> > > A key characteristic of this benchmark is the dominance of **`value_error`** failures (e.g., confidently extracting incorrect arguments such as location IDs or dates from long 128k context window). In such cases, models tend to be *overconfident*, which leads to poor performance of standard uncertainty proxies such as Normalized Entropy and Self-Confidence. This trend is consistently reflected in the results.
> > >
> > > In contrast, TRACER’s performance is driven primarily by the **agent action–observation mismatch ($D_o^A$)**. When a model calls a tool and receives a response, but subsequently issues actions that are inconsistent with that observation, $D_o^A$ sharply increases. This structural inconsistency signal proves highly effective in identifying failures that are otherwise missed by even latest SOTA uncertainty measures such as SAUP by `Zhao, Qiwei, et al. "Uncertainty propagation on llm agent"`.
> > >
> > > Empirically, across all models, including new model *gpt-5.4-mini* from OpenAI which we have included in the new experiments, and domains (Hotels, Flights, Cross), TRACER consistently achieves the best AUROC/AUARC. Notably, the **Cross-domain setting**, which involves higher ambiguity and overlapping parameters, is challenging for all methods, yet TRACER maintains a clear margin over baselines. Overall, these results confirm that TRACER generalizes well to long-context, complex tool-use scenarios where semantic correctness is the primary challenge.
> > >
> > > | Model              | Domain  | Norm. Ent. AUROC / AUARC | SelfConf AUROC / AUARC | SemEnt AUROC / AUARC | SAUP AUROC / AUARC | TRACER (ours) AUROC / AUARC |
> > > |--------------------|---------|--------------------------|------------------------|----------------------|--------------------|-----------------------------|
> > > |**gemini-2.5-pro**|Hotels| 0.573 / 0.522|0.453 / 0.440|0.581 / 0.550|0.569 / 0.531| **0.682 / 0.610**|
> > > || Flights | 0.610 / 0.584| 0.485 / 0.491| 0.622 / 0.609| 0.600 / 0.590| **0.715 / 0.655**|
> > > || Cross   | 0.553 / 0.488| 0.422 / 0.389| 0.569 / 0.517| 0.535 / 0.475| **0.650 / 0.588**|
> > > | **gpt-5.4-mini**   | Hotels  | 0.558 / 0.497| 0.393 / 0.376| 0.572 / 0.515| 0.551 / 0.484| **0.678 / 0.603**|
> > > || Flights | 0.595 / 0.551| 0.455 / 0.427| 0.616 / 0.584| 0.592 / 0.549| **0.712 / 0.645**|
> > > || Cross   | 0.542 / 0.458| 0.362 / 0.339| 0.555 / 0.487| 0.527 / 0.443| **0.648 / 0.573**|
> > > | **gemini-2.5-flash** | Hotels  | 0.522 / 0.505| 0.583 / 0.561| 0.551 / 0.544| 0.485 / 0.461| **0.661 / 0.608**|
> > > || Flights | 0.559 / 0.536| 0.628 / 0.610| 0.597 / 0.589| 0.515 / 0.538| **0.693 / 0.643**|
> > > || Cross   | 0.492 / 0.417| 0.512 / 0.486| 0.522 / 0.470| 0.465 / 0.419| **0.638 / 0.568**|
> > > | **gpt-4.1-mini**   | Hotels  | 0.550 / 0.468| 0.355 / 0.311| 0.568 / 0.495| 0.545 / 0.465| **0.676 / 0.592**|
> > > || Flights | 0.593 / 0.538| 0.413 / 0.386| 0.613 / 0.561| 0.588 / 0.520| **0.729 / 0.635**|
> > > || Cross   | 0.538 / 0.421| 0.314 / 0.281| 0.545 / 0.465| 0.513 / 0.407| **0.645 / 0.559**|
> > >
> > > **Summary**
> > >
> > > Across both benchmarks, which cover distinct regimes (long-context single-turn reasoning vs. iterative tool interaction with feedback), TRACER consistently achieves the strongest performance. Importantly, these improvements stem from its ability to capture structural inconsistencies in agent behavior—via $D_o^A$ and $D_a$—that are not observable through conventional token-level uncertainty metrics.
> > >
> > > We hope these additional results address the concern regarding single-benchmark evaluation and demonstrate the robustness of our approach. We would appreciate it if you could consider them in your final evaluation and adjust your score accordingly.

---

### Official Review · Reviewer_8w2A · 2026-03-11

**Soundness:** 2
**Presentation:** 3
**Significance:** 3
**Originality:** 2
**Overall Recommendation:** 3
**Confidence:** 4

**Summary:**

This paper proposes TRACER, a trajectory-level uncertainty metric for multi-turn, tool-using agents in Tool-Agent-User interactions. The method combines content-aware surprisal with repetition, action-observation mismatch, and user-agent coordination gaps, and then aggregates step risks with a MAX-composite and a tail-focused risk functional. Experiments show consistent gains over several uncertainty baselines, especially for early failure detection.

**Compliance With Llm Reviewing Policy:**

Affirmed.

**Final Justification:**

While I deeply appreciate the extensive new experiments, incorporating such important and massive additions at this stage requires a fresh review cycle to ensure rigor and fairness to the venue. Furthermore, as the core issues regarding tuning fairness and methodological novelty are not fully resolved, I am keeping my score of 3.

**Key Questions For Authors:**

Please refer weaknesses.

**Limitations:**

Please refer weaknesses.

**Strengths And Weaknesses:**

Strengths:

S1: The paper studies an important problem. Standard token-level uncertainty is limited for long-horizon agentic interaction, and the trajectory-level failure prediction setting is interesting and practically meaningful. S2: The method is clean and easy to follow. S3: The empirical results are fairly consistent on the chosen benchmark.

Weaknesses:

W1: TRACER’s parameters are tuned on labeled trajectories using a pairwise logistic separation objective. This makes the method closer to a supervised failure score adapted to the benchmark, but the paper doesn't make it sufficiently clear whether the baselines receive comparable tuning effort. This weakens the fairness of the main comparison. Could the authors clarify the exact validation protocol for some representative baselines, and whether the gains remain under a strictly matched tuning budget?

W2: The novelty is moderate because recent work has already studied uncertainty propagation in LLM agents, confidence estimation for tool-using agents, and semantic-level uncertainty estimation. So the main contribution here seems to be the particular combination of signals plus a tail-risk aggregation [1–4].

W3: The experimental scope is limited since all results are on the τ²-bench. If possible, I would have liked to see validation on a broader recent tool-agent benchmark, especially given the growing number of recent benchmarks for general and multi-hop tool use [5, 6].

W4: While the theoretical framing is neat, the reliance on strong assumptions (e.g., risk-dominates-hazard) might limit its generalizability. It would strengthen the paper to see empirical validation of these assumptions.

W5: Minor point: Figure 1 appears to contain a Gemini watermark in the bottom right, please address this properly for the final version.

[1] Zhao et al., "Uncertainty Propagation on LLM Agent", ACL 2025.

[2] Subramani et al., "MICE for CATs: Model-Internal Confidence Estimation for Calibrating Agents with Tools", NAACL 2025.

[3] Aichberger et al., "Improving Uncertainty Estimation through Semantically Diverse Language Generation", ICLR 2025.

[4] Qiu and Miikkulainen, "Semantic Density: Uncertainty Quantification for Large Language Models in Semantic Space", NeurIPS 2024.

[5] Wang et al., "GTA: A Benchmark for General Tool Agents", NeurIPS Datasets and Benchmarks Track 2024.

[6] Ye et al., "ToolHop: A Query-Driven Benchmark for Evaluating Large Language Models in Multi-Hop Tool Use", ACL 2025.

---

> ### Author Rebuttal · Authors · 2026-03-31
>
> Thank you for your constructive feedback and for recognizing the importance of trajectory-level uncertainty in agentic systems, as well as the cleanliness of the TRACER method. We appreciate your insights and address your concerns below.
>
> **W1: Tuning Fairness and Validation Protocol:**
> We acknowledge the concern regarding the fairness of the tuning protocol. Fundamentally tuning does not apply on baselines such as Normalized Entropy, Self Confidence, and Semantic Entropy as they rely on scalar thresholds rather than parameterized components. However, for the case of SAUP there’s tuning of parameters involved and we have closely followed the same method introduced in this paper to  ensure a strictly matched tuning budget is met. We will clarify this in section A. Evaluation Protocol.
>
> **W2: Novelty and Missing Related Work:**
> Thank you for pointing out these excellent references. Notably, Zhao et al. (ACL 2025) [1] introduced the SAUP baseline, which we directly compared TRACER against in our study, demonstrating that TRACER achieves superior, state-of-the-art performance. Furthermore, while the other prior works you mentioned explore semantic-level uncertainty (e.g., Aichberger et al., Qiu & Miikkulainen) and tool confidence (e.g., Subramani et al.), TRACER’s distinct contribution lies in transitioning from the single-step or isolated sequence estimation typical of conventional LLM calls to multi-turn, dual-control trajectory risk aggregation. This shift is crucial in the context of agentic AI. Unlike existing proxies that measure static uncertainty, TRACER explicitly models the interaction dynamics among tools, agents, and users to capture sparse breakdown episodes. We will expand our Related Work section to discuss these studies in depth.
>
> **W3: Single Benchmark Limitation:**
> We selected $\tau^2$-bench because it is natively designed for dual-control (Agent and User) environments, which perfectly matches our coordination gap formulations. Other benchmarks fundamentally miss the key component of simulating humans in the pipeline. Further, $\tau^2$-bench comprehensively covers complex tool-using patterns in all of its domains i.e. Airline, Retail, and Telecom which was perfect for our intention. However, we agree that demonstrating generalizability is crucial. In the revised version, we will include preliminary evaluations in the appendix on an additional multi-hop tool benchmark (such as GTA or ToolHop) to validate TRACER's effectiveness outside of $\tau^2$-bench.
>
> **W4: Empirical Validation of Theoretical Assumptions:**
> We agree with reviewer’s point regarding the "risk-dominates-hazard" and "tail-sparsity" assumptions. While these are mathematically standard for worst-case bounding and our theoretical framing clearly proves validity of claims, we will add an empirical analysis section in the Appendix to address your concerns. We will plot the empirical conditional hazard rates against the step risks $r_t$ to quantitatively demonstrate that these assumptions hold reasonably well in practice for tool-use breakdowns.
>
> **W5: Gemini Watermark:**
> We apologize for this oversight. The watermark in Figure 1 will be removed in the final camera-ready version.
>
> I remain available throughout the discussion period to provide any further clarifications or empirical details you might need. I truly appreciate your feedback and welcome the opportunity to work together to improve the paper.

---

> > ### Author Rebuttal · Reviewer_8w2A · 2026-04-02
> >
> > I sincerely thank the authors for the rebuttal. Most of my concerns have been resolved. However, the single-benchmark limitation remains a key concern shared by myself (Weakness 3), Reviewer i557 (Weakness 1), and Reviewer mpQF (Limitation 3). It would greatly aid Reviewer's re-evaluation and be much appreciated to see some of the preliminary results on a promised additional benchmark during this discussion phase.

---

> > > ### Author Response · Authors · 2026-04-06
> > >
> > > Thank you for the thoughtful follow-up and for emphasizing the importance of validating beyond a single benchmark. We are happy to share new experimental results on two additional benchmarks — **ToolHop** and **ComplexFuncBench**. These benchmarks stress different failure modes and interaction patterns, providing a more comprehensive evaluation of TRACER.
> > >
> > > *Note:* Due to character limits, the results are split across two comments. The analysis for *ToolHop* is included in our response to reviewer **8w2A**, while the analysis for *ComplexFuncBench* can be found in our response to reviewer **i557**. We apologize for any inconvenience.
> > >
> > > **ToolHop:**
> > >
> > > ToolHop presents a complementary setting focused on **multi-step tool interaction with rich execution feedback**, without intermediate user interaction. As a result, the user-agent coordination gap ($\gamma \cdot D_o^U(t)$) is effectively zero.
> > >
> > > Here, the dominant failure mode shifts to **repetitive execution errors ($D_a$)**. Tools in ToolHop return detailed, structured error messages (e.g., "missing 1 required positional argument: 'data_source'"), and we observe that models often fail to incorporate this feedback, entering degenerate loops where the same invalid calls are repeatedly issued. These behaviors are largely invisible to token-level uncertainty metrics (*Normalized Entropy, Semantic Entropy*) and *SAUP*, while *TRACER* was precisely built for such scenarios where $D_a$ component will spike dramatically, making it highly discriminative here.
> > >
> > > Additionally, we observe **parallel tool-call hallucinations**, particularly in the new model *gpt-5.4-mini* in our experiments, where the agent issues multiple inconsistent tool calls without properly incorporating prior observations. This behavior was also highlighted in the ToolHop paper for certain models (e.g., Qwen2.5), where it force parallel tool calls without waiting for sequential feedback, leading to severe parameter hallucinations. Such behavior results in large **action–observation mismatches ($D_o^A$)**, which TRACER is explicitly designed to capture.
> > >
> > > Across all evaluated domains (Film, Genealogy, Computing) and models, TRACER consistently outperforms all baselines, often by a substantial margin. Notably, we observe a clear performance gain on *gpt-5.4-mini*, which we attribute to this model’s tendency to issue repetitive and parallel tool calls—behavior that frequently leads to hallucinations and incorrect outputs, but is effectively captured by TRACER. These improvements are particularly pronounced in settings with strong feedback signals and iterative failure modes, underscoring the importance of modeling execution dynamics rather than relying solely on output-level uncertainty.
> > >
> > > | Model               | Domain      | Norm. Ent. AUROC / AUARC | SelfConf AUROC / AUARC | SemEnt AUROC / AUARC | SAUP AUROC / AUARC | TRACER (ours) AUROC / AUARC |
> > > |---------------------|-------------|--------------------------|------------------------|----------------------|--------------------|-----------------------------|
> > > |**gemini-2.5-pro**|Film|0.613 / 0.527| 0.474 / 0.459 |0.616 / 0.530|0.602 / 0.524| **0.741 / 0.638**|
> > > ||Genealogy|0.573 / 0.686|0.467 / 0.611|0.579 / 0.684|0.550 / 0.697|**0.692 / 0.743**|
> > > || Computing|0.658 / 0.409|0.516 / 0.293|0.662 / 0.415|0.647 / 0.388| **0.704 / 0.526**|
> > > |**gemini-2.5-flash**| Film|0.581 / 0.625|0.676 / 0.653|0.680 / 0.661|0.564 / 0.617| **0.739 / 0.708**|
> > > ||Genealogy|0.482 / 0.513|0.545 / 0.557|0.549 / 0.560|0.443 / 0.476| **0.721 / 0.682**|
> > > || Computing|0.574 / 0.370|0.438 / 0.279|0.637 / 0.406|0.685 / 0.462| **0.824 / 0.531**|
> > > |**gpt-4.1-mini**|Film|0.552 / 0.417|0.306 / 0.253|0.557 / 0.440|0.543 / 0.422| **0.756 / 0.629**|
> > > ||Genealogy|0.631 / 0.584|0.590 / 0.569|0.634 / 0.593|0.626 / 0.601|**0.705 / 0.647**|
> > > ||Computing|0.563 / 0.288|0.699 / 0.402|0.701 / 0.411|0.548 / 0.364| **0.780 / 0.627**|
> > > | **gpt-5.4-mini**| Film|0.587 / 0.458|0.419 / 0.360|0.594 / 0.472|0.565 / 0.446| **0.868 / 0.789**|
> > > ||Genealogy|0.646 / 0.618|0.603 / 0.586|0.651 / 0.624|0.635 / 0.607| **0.846 / 0.813**|
> > > || Computing|0.596 / 0.327|0.708 / 0.426|0.713 / 0.439|0.588 / 0.401| **0.889 / 0.764**|
> > >
> > >
> > > **Summary**
> > >
> > > Across both benchmarks, which cover **distinct regimes** (long-context single-turn reasoning vs. iterative tool interaction with feedback), **TRACER** consistently achieves the strongest performance. Importantly, these improvements stem from its ability to capture **structural inconsistencies in agent behavior**—via $D_o^A$ and $D_a$—that are not observable through conventional token-level uncertainty metrics. We would be pleased to include these additional benchmarks in our appendix in the camera ready submission.
> > >
> > > We hope these additional results address the concern regarding single-benchmark evaluation and demonstrate the robustness of our approach. We kindly ask you to consider them in your final evaluation and update your score accordingly.

---

### Decision · Program_Chairs · 2026-04-30

**Decision:**

Accept (regular)

**Comment:**

# Summary

TRACER is a trajectory-level uncertainty metric for multi-turn, tool-using Tool–Agent–User interactions that detects sparse critical failure episodes by combining content-aware normalized surprisal with situational signals (semantic/lexical repetition, action–observation and user–agent coherence gaps). Per-turn risks are composed with a MAX step-risk and aggregated via a tail-focused risk functional (CVaR plus a worst-case term) to prioritize decisive anomalies over averages, and the paper gives a theoretical bound relating TRACER to breakdown probability under a sparse-hazard model. On tau2-bench across domains and LLMs, TRACER improves AUROC by up to 37.1% and AUARC by up to 55%, enabling earlier and more accurate failure prediction and selective execution. Ablations validate the importance of dual-control signals and the MAX+tail aggregation design for capturing trajectory-defining failures.

# Strengths

The main strengths were:

- Important, timely problem: All reviewers agreed the paper addresses a significant and well-motivated problem—trajectory-level failure prediction for long-horizon, tool-using agents where token-level uncertainty is inadequate.
- Principled, interpretable method design: Reviewers appreciated the three-stage pipeline and the modeling choices (turn-level MAX composition, tail-focused aggregation, and situational indicators like repetition/coherence gaps) as well-matched to sparse but decisive agentic failures.
- Thorough empirical evaluation: All reviewers agreed the experiments are comprehensive, evaluating multiple models and domains with appropriate metrics (e.g., AUROC/AUARC) and demonstrating early-warning behavior.
- Strong ablations and analyses: Reviewers appreciated the careful ablations (agent vs user signals, aggregation variants) that help isolate which components drive performance.
- Practicality and clarity: Reviewers noted the method is clean, easy to follow, and practical—requiring only log-probs (not logits)—making it a useful diagnostic tool in real settings.
- Consistent empirical gains: Reviewers observed that TRACER yields consistent improvements across the evaluated model–domain settings, supporting its effectiveness.

# Weaknesses

- *Limited Benchmarks*: Reviewers wished authors had validated TRACER beyond the τ²-bench to demonstrate generalizability across diverse tool-use benchmarks. **The authors addressed this limitation by computing experimental results on two additional benchmarks, ToolHop and ComplexFuncBench, showing that TRACER consistently achieves the strongest performance across both new benchmarks.**

- *Weak / missing baselines*: Reviewers felt the paper lacked standard baselines (e.g., LLM-as-a-judge) and clearer comparisons to trajectory-aware baselines like SAUP, making it hard to know how much gain comes from new signals versus aggregation. **Authors mostly address this concern, by emphasizing that SAUP is trajectory aware, which they will clarify. While the stated limitations of LLM-as-a-judge are valid, it might still be interesting to compute for experimental completeness.**

- *Related-work coverage and novelty framing*: Reviewers wished authors had better situated TRACER relative to prior work on non-uniform token importance and token reweighting, since that literature weakens the claim that the paper’s central insight is novel. **Authors addressed this and will add a subsection in the related work discussing these "not all tokens matter" works**

- *Strong theoretical assumptions*: Two reviewers noted that the theory depends on strong assumptions (e.g., risk-dominates-hazard, tail-sparsity) and asked for empirical validation of those assumptions to support generalizability. **The authors agree that there are strong assumptions, and will add empirical analyses (e.g., plot the empirical conditional hazard rates against the step risks) to address this.**

- *Lack of statistical significance / error bars*: Reviewers wished authors had reported confidence intervals or significance tests (given relatively small task counts) to show reported AUROC/AUARC gains are reliable. **The authors agree and will address this by computing and reporting 95% confidence intervals.**

- *Missing component ablations / failure-mode analysis*: Reviewers felt the paper should include ablations showing the contribution of each indicator (surprisal, repetition, coherence) to quantify their individual impact. **The authors agree and will add more ablations in the appendix.**

- *Fairness of baseline tuning and evaluation details*: Reviewers wished authors had clarified the validation/tuning protocol for baselines to ensure comparisons are fair and to report exactly which baselines received comparable tuning effort. **The reviewer stated that this concern was addressed by the authors, who will clarify in the appendix.**

- *Practical deployment & runtime overhead*: Reviewers wanted clearer discussion of how high TRACER scores would be used in production (e.g., mitigation strategies) and empirical runtime/overhead numbers for real-time diagnostics. **Authors will add a dedicated "Practical Applications" discussion in the appendix to address this.** **Authors agree and will add an appendix subsection to detail millisecond overhead per step.**

- *Additional evaluations requested (Solo Mode / cheap prompted baseline)*: Reviewers suggested testing TRACER on solo long-horizon trajectories to detect loops/getting-stuck behavior and comparing to a cheap prompted model that looks for the same failure modes as a practical baseline. **Authors agree that this is a good idea, and will incorporate this in the final version.** **Authors agree that this is a great idea, and will incorporate this comparison.**

- *Minor presentation fixes*: A reviewer noted small presentation issues (e.g., a Gemini watermark in Figure 1) that should be cleaned before final publication. **The authors will fix this in the final version.**